# “Polymerization” of Bimerons in Quasi-Two-Dimensional Chiral Magnets with Easy-Plane Anisotropy

**DOI:** 10.3390/nano14060504

**Published:** 2024-03-11

**Authors:** Natsuki Mukai, Andrey O. Leonov

**Affiliations:** 1International Institute for Sustainability with Knotted Chiral Meta Matter, Kagamiyama, Higashihiroshima 739-8511, Hiroshima, Japan; m226886@hiroshima-u.ac.jp; 2Department of Chemistry, Faculty of Science, Hiroshima University Kagamiyama, Higashihiroshima 739-8526, Hiroshima, Japan

**Keywords:** Skyrmion, bimeron, chiral magnets, induced Dzyaloshinskii–Moriya interaction, Skyrme model of pions, 75.30.Kz, 12.39.Dc, 75.70.-i

## Abstract

We re-examine the internal structure of bimerons, which are stabilized in easy-plane chiral magnets and represent coupled states of two merons with the same topological charge |1/2| but with opposite vorticity and the polarity. We find that, in addition to the vortices and antivortices, bimerons feature circular regions which are located behind the anti-vortices and bear the rotational sense opposite to the rotational sense chosen by the Dzyaloshinskii–Moriya interaction. In an attempt to eliminate these wrong-twist regions with an excess of positive energy density, bimerons assemble into chains, and as such exhibit an attracting interaction potential. As an alternative to chains, we demonstrate the existence of ring-shaped bimeron clusters of several varieties. In some rings, bimeron dipoles are oriented along the circle and swirl clockwise and/or counterclockwise (dubbed “roundabouts”). Moreover, a central meron encircled by the outer bimerons may possess either positive or negative polarity. In other rings, the bimeron dipoles point towards the center of a ring and consequently couple to the central meron (dubbed “crossings”). We point out that the ringlike solutions for baryons obtained within the Skyrme model of pions, although driven by the same tendency of the energy reduction, yield only one type of bimeron rings. The conditions of stability applied to the described bimeron rings are additionally extended to bimeron networks when bimerons fill the whole space of two-dimensional samples and exhibit combinations of rings and chains dispersed with different spatial density (dubbed bimeron “polymers”). In particular, bimeron crystals with hexagonal and the square bimeron orderings are possible when the sides of the unit cells represent chains of bimerons joined in intersections with three or four bimerons, respectively; otherwise, bimeron networks represent disordered bimeron structures. Moreover, we scrutinize the inter-transformations between hexagonal Skyrmion lattices and disordered bimeron polymers occuring via nucleation and mutual annihilation of merons within the cell boundaries. Our theory provides clear directions for experimental studies of bimeron orderings in different condensed-matter systems with quasi-two-dimensional geometries.

## 1. Introduction

Chiral magnetic Skyrmions [1,2] are topological solitons imbedded into homogeneously magnetized states and exhibiting repulsive inter-Skyrmion potentials [3,4]. Their relevant length scale [5,6] can be tuned based on the competition between direct interaction and Dzyaloshinskii–Moriya interaction (DMI) [7,8], and ranges from a few atomic spacings up to microns [9]. Skyrmions were first experimentally identified in bulk cubic helimagnets such as the itinerant magnets MnSi [10] and FeGe [11] and the Mott insulator Cu_2_OSeO_3_ [12], where they represent three-dimensional (3D) filaments along the field direction [13,14]. Afterwards, 3D isolated Skyrmions (IS) [15,16] have been microscopically spotted in thin layers of the cubic helimagnets (Fe,Co)Si [17] and FeGe [18], where they undergo an additional screw towards the confining surfaces and thereby gain stability in a broad range of temperatures and magnetic fields [19].

Truly two-dimensional (2D) Skyrmions are stabilized, e.g., in bulk polar magnets with $C_{nv}$ symmetry, such as GaV_4_S_8_ and GaV_4_Se_8_ [20,21] (see the exact phenomenological form of DMIs in [1] for chiral magnets with different crystal symmetries). In these *Néel* skyrmions, the magnetization rotates radially from the Skyrmion center to the outskirt, as shown in Figure 1a. Alternatively, thin-film multilayer structures represent a 2D arena for *Néel* Skyrmions, where they can be manipulated as particle-like entities. The breaking of the inversion symmetry and the resulting DMI both originate from the interfaces between a heavy metal layer and Skyrmion-hosting magnetic layer, such as occurs in PdFe/Ir (111) bilayers [22]. Such systems are extremely versatile as regards the choice of the magnetic, non-magnetic, and capping layers as well as the possibility of stacking.

Recently, Skyrmions have generated enormous interest due to the prospect of their applications in information storage and processing [23,24,25]. Skyrmions are topological solitons [26], have the nanometer size [9], and can be manipulated by electric currents [27,28]. The static and dynamic properties of Skyrmions and their interactions with quenched disorder and pinning are reviewed in [29].

The field configuration of isolated Skyrmions can be characterized by the topological charge or Skyrmion number *Q*, which arises in the map from the physical 2D space to the target space $S_{2}$ [2]. The topological charge describes how many times the magnetic moments wrap around a unit sphere in the mapping [30]. In spherical coordinates for the magnetization, m=(sinθ(ρ)cosψ(φ),sinθ(ρ)sinψ(φ),cosθ(ρ)), and in cylindrical coordinates for the spatial coordinates, r=(ρcosφ,ρsinφ); thus, the expression for the topological charge becomes particularly simple:(1)$$\begin{array}{cc} \; & Q=\frac{1}{4\pi }\int m\cdot \left(\frac{\partial m}{\partial x}\times \frac{\partial m}{\partial y}\right)dxdy=\\ \; & \frac{1}{4\pi }\int_{\theta_{1}}^{\theta_{2}}\sin\theta d\theta \int_{\varphi_{1}}^{\varphi_{2}}md\varphi =\frac{1}{4\pi }m\left(\varphi_{1}-\varphi_{2}\right){\cos\theta |}_{\theta_{1}}^{\theta_{2}} \end{array}$$Here, ψ=mφ+γ, *m* is the vorticity, and γ is the the helicity.

In particular, for ordinary isolated Néel Skyrmions,
(2)$$\gamma =0;\pi ,\;m=1,\;\theta_{1}=\pi ,\;\theta_{2}=0,\;\varphi_{1}=0,\;\varphi_{2}=2\pi ;$$
therefore, Q=−1. The helicity does not explicitly enter into the final formula for the topological charge (Equation 1).

Remarkably, the above expression (Equation 1) allows two types of merons to be introduced in a systematic way, and these merons possess fractional topological charges ±1/2 (henceforth called merons and anti-merons).

### 1.1. Bimerons within the Spiral States of Chiral Magnets

In the first type of merons, while the polar angle of the magnetization makes the full swing from π to 0, the azimuthal angle only allows half of the $S_{2}$ sphere to wrap (Figure 1b):(3)$$\begin{array}{cc} \; & m=1,\;\theta_{1}=\pi ,\;\theta_{2}=0,\;\varphi_{1}=0,\;\varphi_{2}=\pi ,\\ \; & Q=\left(1/4\right)\left(-\cos\theta {|}_{\pi }^{0}\right)=-1/2. \end{array}$$

Bimerons, as coupled states of such anti-merons with total charge Q=−1, serve as quanta of the phase transition between the cycloid and the hexagonal Skyrmion lattice (SkL) [31], i.e., they are visualized as ruptures of the cycloidal state with the wave vector q (Figure 1c, dashed blue circle). The energy of such an anti-meron pair becomes negative with respect to the cycloidal background at some critical field, which underlies the avalanche-like cycloid-SkL transition (see [32] for details) observed experimentally, e.g., in thin-film helimagnets [17,18]. In the phase diagram of states (Figure 2a), these bimerons are formed above the c−d line when their energy becomes negative with respect to the spiral state.

Merons with Q=+1/2 are formed for
(4)$$m=1,\;\theta_{1}=0,\;\theta_{2}=\pi ,\;\varphi_{1}=0,\;\varphi_{2}=\pi ,$$
and constitute bimerons with Q=+1 (Figure 1c, dashed red circle). Such bimerons are counterparts of the corresponding Néel Skyrmions with positive polarity. Remarkably, it is possible to stabilize bimerons with Q=0 by coupling merons and anti-merons (Figure 1c). In Figure 1c, the meron and the anti-meron in the neighboring cycloidal periods may move towards each other and annihilate, leaving the remaining meron and the anti-meron pair coupled [32]. Such (Q=0)-bimerons have been shown to experience no Skyrmion Hall effect and to move straight along the current, making them promising information carriers for spintronic devices [32].

### 1.2. Bimerons within the Tilted Ferromagnetic States

In the second type of bimerons, the magnetization rotates only to the equatorial plane, whereas the azimuthal angle makes the full turn. Then, merons with Q=+1/2 are obtained (first row in Figure 1e) for
(5)$$\begin{array}{cc} \; & m=1,\;\theta_{1}=0,\;\theta_{2}=\pi /2,\;\varphi_{1}=0,\;\varphi_{2}=2\pi ,\\ \; & m=-1,\;\theta_{1}=\pi ,\;\theta_{2}=\pi /2,\;\varphi_{1}=0,\;\varphi_{2}=2\pi , \end{array}$$
while anti-merons with Q=−1/2 are obtained (second row in Figure 1e) for
(6)$$\begin{array}{cc} \; & m=1,\;\theta_{1}=\pi ,\;\theta_{2}=\pi /2,\;\varphi_{1}=0,\;\varphi_{2}=2\pi ,\\ \; & m=-1,\;\theta_{1}=0,\;\theta_{2}=\pi /2,\;\varphi_{1}=0,\;\varphi_{2}=2\pi . \end{array}$$

Experimentally, isolated merons have been observed and investigated in confined magnetic disks [33,34], as circular geometry is compatible with the boundary conditions.

Bimerons, which fit into the in-plane state of the magnetization (the red-shaded region in the phase diagram in Figure 2a) can be made only out of merons with the same sign of the topological charge (Figure 1f) and opposite vorticity *m*, i.e., two bimeron varieties are possible (with the magnetization within circular merons pointing either up or down). Identical merons with the same vorticity would inevitably create an antivortex in the interstitial region and become a trimeron, whereas merons with the opposite topological charges would mutually annihilate.

An intuitive model for the bimeron states in Figure 1f can be obtained by rotating all of the magnetization vectors within an ordinary Néel Skyrmion (the middle spin distribution in (f)) by the angle π/2 either clockwise or counterclockwise around the *y* axis. In this way, the magnetization in the center of the IS (point *O*) now points horizontally along the dipole moment of the bimeron. Two points with $m_{z}=0$ within the IS now become the centers of circular and crescent-shaped merons (points *A* and *B* in Figure 1f). This construction can be used as an initial configuration for numerical simulations; however, the antivortex part becomes crescent-shaped after the relaxation. For frustrated bimerons stabilized by the competing exchange interactions such an initial state is very close to the real solution, as vortices and antivortices are energetically degenerate (for details, see [35]).

Bimerons have been theoretically predicted in different condensed-matter systems, including frustrated magnets [35], antiferromagnets [36], and helimagnets with $C_{nv}$ crystal symmetry [37], and have been observed in epitaxial antiferromagnetic film [38] and multilayered FM film [39] upon external fields. The on-demand creation of bimerons in a cubic helimagnet Fe_0.5_Co_0.5_Ge was recently reported in [40]. Chains of bimerons dubbed “schools” were observed in thin layers of chiral liquid crystals [41] with thickness slightly smaller than the spiral pitch. Bimerons have been predicted by first-principle calculations and atomistic simulations as applied to van der Waals magnetoelectric CrISe/In_2_Se_3_ heterostructures [42]. Moreover, merons have been introduced in the context of quark confinement in the nonlinear O(3) σ model [43,44]. A loosely bound collection of parallel Skyrmion chains was demonstrated in nematic superconductors for low fields in [45]. Such nematic systems are two-component superconductors that break rotational symmetry but exhibit a mixed symmetry that couples spatial rotations and phase difference rotations. The chiral p-wave superconducting state supports a rich spectrum of topological excitations different from those in conventional superconducting states [46]. With the appropriate sign of the phase winding, two-quanta vortices were shown to always be energetically preferred over two isolated single quanta vortices.

In the present manuscript, we focus on the particular type of bimerons stabilized in an easy-plane magnet with the Dzyaloshinskii–Moriya interaction [47,48]. Indeed, the bimerons stabilized by the in-plane magnetic field already exhibit entirely different properties [49].

Anisotropy-shaped bimerons can become suitable for practical applications, as they do not, for example, require applied magnetic fields and exist in two varieties with opposite polarity (see the red-shaded region of the phase diagram in Figure 2a). The dynamic switching of bimerons holds potential in regard to qubits for quantum computing [50]. Current-driven motion of bimerons has been shown to exhibit two distinct time regimes: bimerons initially rotate towards the current direction, then subsequently move along the current [51]. Exploiting the fact that bimerons of opposite topological charges may exist in the same material, a bimeron pair can be generated. A topologically trivial vortex–antivortex pair was shown to be created out of fluctuations of the polarized state in [43].

In the present paper, we first revisit the internal properties of bimerons stabilized in quasi-2D chiral magnets with the easy-plane anisotropy (EPA). We show that bimerons possess parts of their magnetization distribution with the reverse rotational sense, i.e., opposite to the sign of the DMI. This phenomenon has been overlooked thus far, the reason presumably being the small numerical grids used in previous simulations. For example, the bimeron solutions in [43] are distorted and do not fit perfectly into the unit cell; the magnetization at the boundaries is different from the in-plane state.

As a result of the positive energy density associated with the wrong rotational sense, bimerons develop attracting interaction and align into chains. As an alternative, round bimeron clusters can be created in an attempt to extinguish all parts with the wrong twist. We perform an exhaustive analysis of different bimeron clusters with closed structures depending on the number of constituent bimerons. We investigate the clusters with opposite polarities of the central merons as well as with bimeron dipoles either arranged along the circle or pointing toward the center. We touch on the problem of their internal stability and inter-transformation between different cluster varieties. Bimeron polymers filling the whole space are shown to comply with the principles of inter-bimeron coupling. We additionally discuss the interconnection between hexagonal Skyrmion lattices and bimeron polymers.

## 2. Phenomenological Theory of Bimerons in Two-Dimensional Helimagnets

The magnetic energy density of a non-centrosymmetric ferromagnet with $C_{nv}$ symmetry (or with induced DMI, which has the same functional form) can be written as the sum of the exchange, DMI, Zeeman, and anisotropy energy density contributions, correspondingly:(7)$$w\left(m\right)=\sum_{i,j}{\left(\partial_{i}m_{j}\right)}^{2}+w_{DMI}-m\cdot h-k_{u}m_{z}^{2}.$$

We neglect the influence of dipole–dipole interactions due to the magnetic charges formed within different states with Néel-like type magnetization rotation. We assume that the DM interactions suppress demagnetization effects and are the main driving force leading to the magnetization rotation and equilibrium periodicity. Moreover, the shape anisotropy in this case represents an additional correction of the easy-plane anisotropy. The influence of dipole–dipole interactions on the effects found in the present manuscript will be considered elsewhere.

Here, we introduce the nondimensional units to make the results more encompassing and to allow their direct mapping to any material system. Spatial coordinates x are measured in units of the characteristic length of modulated states $L_{D}$. The value $\lambda =4\pi L_{D}$ for zero magnetic field is the *period of the cycloid*. A>0 is the exchange stiffness, *D* is the Dzyaloshinskii constant, and $k_{u}$ is the non-dimensional anisotropy constant, which leads to the easy-plane magnetization, i.e., $k_{u}<0$. If one needs to use the results of the simulations for a specific material system, one can easily calculate these non-dimensional units and find the parameter point on the phase diagram in Figure 2a.
(8)$$L_{D}=A/D,k_{u}=K_{u}M^{2}A/D^{2},h=H/H_{0},H_{0}=D^{2}/A|M|,m\left(x,y\right)=M/\left|M\right|$$

In the above, h is the magnetic field applied along the *z* axis. The magnetization vector m(x,y) is normalized to unity. The DMI energy density has the following form specific for chiral magnets with the $C_{nv}$ symmetry:(9)$$w_{DMI}=m_{x}\partial_{x}m_{z}-m_{z}\partial_{x}m_{x}+m_{y}\partial_{y}m_{z}-m_{z}\partial_{y}m_{y}$$
where $\partial_{x}=\partial /\partial x,\;\partial_{y}=\partial /\partial y$.

In the following, we consider a 2D film of a ferromagnetic material on the xy plane using free boundary conditions, which allows us to address different circular structures composed of bimerons that are incompatible with the unidirectional in-plane magnetization. Alternatively, we may use periodic boundary conditions along the polar axis to address bulk helimagnets with $C_{nv}$ symmetry. All bimeron structures are minimized on the grid 4096×4096 with cell size 0.1. The inhomogeneous magnetization distributions formed at the sample edges in this case have no impact on the magnetization distributions in the center of the film where all bimeron clusters are placed.

For Model (Equation 7), only modulated one-dimensional (1D) phases (cycloids and elliptical cones, Figure 2b,c) and two-dimensional (2D) phases (Skyrmions) with propagation directions perpendicular to the polar axis, i.e., in the xy-plane, are energetically favored. The phase diagram of states for Model (Equation 7) is shown in Figure 2a [37,51]. In the following, we avoid the regions of the phase diagrams with these modulated states, i.e., we impose the restriction on the anisotropy value as $k_{u}<-1$. Bimerons are embedded into the tilted ferromagnetic phase (TFM, Figure 2e) with the polar angle θ (the red-shaded region in Figure 2a):(10)$$\theta_{TFM}=\arccos\left(h/2k_{u}\right),$$
i.e., for $h=2k_{u}$ the magnetization is saturated along the field (Figure 2d) and all bimerons transform into ordinary Skyrmions. For h=0, the magnetization points in the in-plane direction. The azimuthal angle of the TFM is constant for isolated bimerons as well as for their chains, although it is a function $\psi_{TFM}\left(x,y\right)$ for numerous bimeron ensembles, which may reach a simple form $\psi_{TFM}\left(\varphi \right)$ for bimeron clusters with circular symmetry.

As our primary numerical tool for minimizing the functional (Equation 7), we use the MuMax3 software package (version 3.10), which calculates the magnetization dynamics by solving the Landau–Lifshitz equation using the finite difference discretization technique [52,53]. To double-check the validity of the obtained solutions, we use our own numerical routines as well, which are explicitly described in, e.g., [54], and are reproduced here for convenience.

### Energy Minimization

For rigorous minimization of the functional (Equation 7), the Euler–Lagrange equations are nonlinear partial differential equations. These equations have been solved via the numerical energy minimization procedure using finite-difference discretization on grids with adjustable grid spacings and free boundary conditions [54]. The components of the magnetization vector m were evaluated in the knots of the grid, and for calculation of the energy density (Equation 7) we used finite-difference approximation of derivatives with different precision up to eight points as neighbors. To check the stability of the numerical routines, we additionally refined and coarsened the grids. For the axial fields, we used grid spacings $\Delta_{y}\approx \Delta_{x}$ to ensure that the grids were approximately square in the xy plane to reduce the artificial anisotropy incurred by the discretization. The final equilibrium structure for the two-dimensional modulated states was obtained according to the following iterative procedure of the energy minimization using simulated annealing and a single-step Monte Carlo dynamics with the Metropolis algorithm:

(i) The initial configuration of magnetization vectors in the grid knots for Monte Carlo annealing was chosen appropriately to ensure relaxation to a desired particle-like state. As initial states for the different bimeron clusters obtained in the present paper, we used bimeron cores and arranged them into a desired texture. In this way, the relaxation procedures lead to the stable spin configuration.

(ii) A point $\left(x_{n},y_{n}\right)$ on a grid is chosen randomly, then the magnetization vector in that point is rotated without changing its length. If the energy change $\Delta H_{k}$ associated with such a rotation is negative, then the new orientation is kept.

(iii) However, if the new state has an energy higher than the last one, then it is accepted probabilistically. The probability *P* depends upon the energy and a kinetic cycle temperature $T_{k}$:(11)$$P=\exp\left[-\frac{\Delta H_{k}}{k_{B}T_{k}}\right],$$
where $k_{B}$ is the Boltzmann constant. Together with the probability *P*, a random number $R_{k}\in \left[0,1\right]$ is generated. If $R_{k}<P$, the new configuration is accepted; otherwise, it is discarded. Generally speaking, at high temperatures $T_{k}$ many states will be accepted, while at lower temperatures the majority of these probabilistic moves will be rejected. Therefore, it is necessary has to choose an appropriate starting temperature for the heating cycles.

(iv) The characteristic spacings $\Delta_{x}$ and $\Delta_{y}$ are adjusted to promote energy relaxation. The procedure is stopped when no further reduction of energy is observed.

## 3. The Properties of Isolated Bimerons

### 3.1. Internal Structure of Bimerons

The internal structure of bimerons and their field- and anisotropy-driven behavior are characterized by a variety of characteristics. As implied by the distribution of $m_{z}$ on the plane xy for h=0 (Figure 1f), bimerons consist of a circular meron with either polarity along with a crescent-shaped meron, which makes it possible for them to fit smoothly into the in-plane state with, e.g., $\psi_{TFM}=0$ or π and to form a localized particle. From the right to the left side of the depicted bimeron (Figure 3a), the magnetization rotates directly from the state θ=π in the center of the circular meron to the state θ=0 in the center of the crescent-shaped meron. The distance between the anti-meron centers is characterized by the dipole moment *p*; we note, however, that the topological charges of both constituent parts are the same, Q=−1/2.

After “zooming” the bimeron profile in the range $m_{z}\in \left[-0.2,0.2\right]$ (inset (i) of Figure 3b), the bimeron contains a region where the magnetization rotation is not supported by the DMI. The center of this region is characterized by the parameter $p^{\prime }$. The cross-section of the magnetization profile along *x* shows that, starting from the crescent center, the magnetization does not rotate immediately to the value θ=π/2 corresponding to the in-plane magnetization; rather, it exceeds this value by an amount $\Delta m_{z}$ and only then rotates back to the in-plane state (inset (ii) of Figure 3b). In this sense, the formation of the area with the opposite rotational sense resembles the structure of the merons confined within nanodiscs with induced DMI investigated in [34]. The magnetization, bearing a local chirality favored by the Dzyaloshinskii–Moriya interactions in the meron core, is encircled by a ring with reverse magnetization rotation [34].

Such peculiar rotational behavior results in distinct energy characteristics of bimerons. Figure 3c shows the DMI energy distribution on the xy plane. The circular region encircled by the dotted white line has positive energy density, as do the two wing-shaped regions. The rest of the magnetization pattern bears negative rotational energy. The same energetic peculiarities are reflected in the color plots for the total energy density (Figure 3d).

Remarkably, the bimeron does not exhibit any region with negative energy density, which makes it drastically different from its counterpart realized for the easy-axis anisotropy; within an isolated Néel Skyrmion, the energy density is positive in the circular region around the center, whereas it becomes negative in the extended ring-shaped area stretching up to infinity (for details, see, e.g., [55]). In the same way, the DMI energy density within modulated states for the easy-axis case appears with a negative magnitude. This distinction, in particular, poses a problem with regard to the mechanism of bimeron condensation into a bimeron lattice. The topological charge density has only regions with the negative sign, which contribute to the total charge Q=−1 (Figure 3e). The xy color plots for the exchange and anisotropy energy densities are shown in Figure 3f,g. These graphs are plotted for completeness, and are not analyzed further.

The field-dependent characteristics of the (Q=−1)-bimeron demonstrate its gradual transformation into an isolated Skyrmion (Figure 4). Indeed, with the increasing field, the homogeneous background surrounding the bimeron gradually tilts towards the field according to (Equation 10). This process is accompanied by the circular area moving away from the main particle (the red curve $p^{\prime }$ in Figure 4a tends to infinity at $h=2k_{u}$) as well as by the increase of the dipole moment itself (the blue curve $p^{\prime }$ in Figure 4a; and see the ratio $p^{\prime }/p$ in (c)). The increasing ratio of $p^{\prime }/p$ indicates that the aforementioned process occurs faster for the circular area of the wrong twist (Figure 4c).

The “depth” $\Delta m_{z}$ of the area with the wrong rotational sense “shallows” to zero (Figure 4c), as ordinary isolated Skyrmions do not possess any part with the “wrong” twist.

The anisotropy-dependent characteristics, on the contrary, show gradual bimeron shrinkage (Figure 4b,d); *p* and $p^{\prime }$ tend to zero for stronger anisotropy values (Figure 4d), with their ratio increasing during this process as well (the lower panel in Figure 4d). $\Delta m_{z}$ turns to zero (upper panel Figure 4d), which now is related to the decreasing bimeron size. The snapshots of the magnetization patterns shown in Figure 4e,f illustrate bimeron evolution during the aforementioned processes. Thus, the influence of the anisotropy on the bimeron size becomes paramount.

The field-dependent characteristics of (Q=+1)-bimerons exhibit their gradual transformation into Skyrmions with the opposite polarity, which are incompatible with the homogeneous background along the field. Therefore, in the following we concentrate only on the behavior of (Q=−1)-bimerons.

### 3.2. Bimeron–Bimeron Attraction

Figure 5a shows the interaction potentials Φ for identical bimerons arranged either head-to-tail (blue curve) or side-by-side (red curve). Here, *d* is the distance between the centers of circular merons. The interaction potential is calculated as the total energy with the energy density (Equation 7) when the centers of circular merons are pinned at particular distances *d*. If the pinning is released, two bimerons move towards each other and accommodate the spin distribution according to the minimum. Note that for the sake of the reproducibility of our results we did not subtract the energy corresponding to two isolated bimerons at large distances *d*.

Although for large inter-bimeron distances both potentials converge to the energy of two non-interacting bimerons, at smaller distances the behavior is drastically different. At point *A* (Figure 5a), the circular region of the second bimeron is covered by the body part of the first bimeron and their mutual energy decreases as a consequence (right panel in Figure 5b). At point *B*, which is located at the minimum of the interaction potential, two bimerons are perfectly coupled, with one circular area with the “wrong” rotational sense being entirely “erased” (middle panel in Figure 5b). Pushing the bimerons even closer results in pronounced distortions of their magnetization distributions (left panel in Figure 5b), which inevitably leads to the energy increase at point *C*. Thus, in the head-to-tail configuration, bimerons form pairs with fixed inter-bimeron distance, implying the attractive nature of their interaction. At point *D*, the magnetization configuration of two bimerons is characterized by the overlapping wing-shaped regions. This configuration leads only to energy increase (Figure 5c), and reveals the repulsive character of the bimeron interaction. The inter-bimeron region with positive DMI energy density only becomes more pronounced in this case [37].

Figure 6a plots the interaction potentials for bimerons with opposite topological charges and oriented towards each other with either circular or crescent-shaped merons, which turn out to be repulsive in both cases. Interestingly, the same magnetization pattern is obtained by the cross-section of truly 3D solitons—Hopfions. In [56], it was argued that Hopfions can be visualized as the result of swirling of such 2D bimerons around the direction of an applied magnetic field (i.e., along the *y* axis in the present illustration). However, it should be taken into account that this can be done only for cubic helimagnets, as they bear rotational DMI-terms along all axes.

Due to the strong deformations of the circular merons facing the interior, such a bimeron pair exhibits stronger repulsion (red curve) as compared with its counterpart with the crescent-shaped merons facing each other (blue curve). Moreover, the full eradication of the two circular areas with the “wrong” twist in the inter-bimeron region (right panels in Figure 6b,c) only brings on bimeron distortions, which outweigh the energetic advantage of this process. However, because two bimerons bear opposite topological charges, they mutually annihilate when placed side-to-side.

### 3.3. Examples of Inter-Skyrmion Attraction

Under certain circumstances, the described mechanism of bimeron attraction becomes apparent for Skyrmions as well.

In [57], chiral-isolated Skyrmions were analysed close to the ordering temperature using the same phenomenological model (Equation 7) with $k_{u}=0$, supplemented with the basic Landau expansion for the homogeneous part of the free-energy. It was shown that the magnetization modulus within Skyrmions remains constant within a broad temperature range and is independent on the value of the applied magnetic field. Near the Curie temperature, however, its longitudinal stiffness decreases and spatial longitudinal modulations of the magnetization become a sizable effect. As a result, isolated Skyrmions develop halos of damped and oscillatory spin twistings with “right” and “wrong” rotational senses of the magnetization. When the magnetization rotates according to the sign of the DMI, the modulus increases, whereas it decreases in the parts with the opposite twists, which possess positive energy density. Due to the strongly oscillatory character of this dependence, two isolated Skyrmions will tend to locate at some discrete equilibrium distances from each other and to be placed in the minima of inter-Skyrmion energy; the overlap of Skyrmion profiles allows the positive energy density to decrease in rings with the “wrong” rotational sense, which represents the same mechanism as described for bimerons.

In frustrated magnets [58,59], the stability of Skyrmions is achieved by the competing ferromagnetic and antiferromagnetic exchange interactions between spins, e.g., on a triangular lattice, which generate higher-order derivative terms in the fundamental continuum form of the magnetic free energy density. The spins at the outskirt of frustrated Skyrmions undergo fan oscillations with decaying amplitude and two rotational senses, which additionally give rise to a number of minima in the Skyrmion–Skyrmion interaction potentials. In the same way, overlapping Skyrmion profiles attempt to reduce the positive energy density in rings associated with rotational sense opposite to that of the Skyrmion.

## 4. Bimeron “Macromolecules”

In the present section, we utilize the most interesting case of attracting bimerons with the same polarity and speculate about the mechanisms of meshing them into clusters with different geometries, a process which we dub “polymerization”. We draw an analogy with the processes occurring in polymer chemistry, in which relatively small molecules called monomers can combine chemically to produce a very large chain-like or network molecule called a polymer. The monomers can be alike or may represent different compounds. In the same way, we can construct macromolecules with two bimeron varieties, using the opposite polarities as building blocks.

### 4.1. Linear Bimeron Macromolecules—Chains

The distinct energy pattern of an isolated bimeron (Figure 2d) implies bimeron chain formation rather than closely packed bimeron clusters with, e.g., hexagonal ordering, as would be the case for ordinary Skyrmions [54]. A bimeron chain runs along the in-plane (or canted) magnetization component of the TFM phase. It can contain any number of bimerons, and each added bimeron erases the circular region with the wrong twist of the preceding chain member [60]. The characteristic color plots for a chain with N=6 bimerons are depicted in Figure 7a–e. Figure 7a shows the color plot for the $m_{z}$-component of the magnetization on the xy plane. Figure 7b shows the energy density of the DMI zoomed in the range [−0.01;0.01]. Figure 7c shows the total energy density zoomed in the range [0;0.1]. Figure 7d shows the energy density of the easy-plane anisotropy (EPA) zoomed in the range [0;0.05]. Figure 7e shows the exchange energy density zoomed in the range [0;0.05].

We note, however, that one circular area with the “wrong” rotational sense of the first bimeron always remains and makes the first pair of bimerons distinguishable from the bimerons within the core of the chain. Moreover, the last bimeron in a chain is slightly different due to the lack of a neighbor. Thus, the energetic advantage of a chain rests on the “perfect” coupling among bimerons reaching the value ΔΦ except for the first bimeron in a chain.

### 4.2. Ring-Shaped Bimeron Macromolecules—“(±)Roundabouts”

The unavoidable remaining region with the wrong rotational sense prompts a chain to swirl into a ring, and consequently to couple the leading bimeron with the last one (Figure 8, Figure 9 and Figure 10). The dynamics of such a process require a more thorough examination, which will be done elsewhere; however, the increasing angle between the neighboring dipoles brings into question the energetic advantage of bimeron coupling. On the other hand, the circular geometry of the samples with smaller radii may naturally instigate such bimeron swirling.

In rings, the energetic advantage due to the absence of circular regions is counterbalanced by the following energetic penalties: (i) the angle between the dipole moments of the neighboring bimerons acquires the value π−2π/N, meaning that the minimum ΔΦ of the interaction potential (Figure 3a) becomes shallower and makes the bimeron coupling less effective, which is especially crucial for small numbers of bimerons in a ring; (ii) the central meron arising in the center of a ring repulses surrounding bimerons, which leads to an energy increase; in particular, the DMI energy density in the region between bimerons and the central meron is larger than that within the bimerons themselves (Figure 8b, Figure 9b and Figure 10b). Certain parts with positive DMI energy remain preserved within the remainders of the bimeron “wings” (the white lobes in Figure 8b).

Interestingly, rings with the opposite circling of bimerons are energetically degenerate, i.e., a ring with counterclockwise (CCW) bimeron swirling (Figure 8) possesses the same total energy as a ring with clockwise (CW) rotation of bimerons (Figure 9). The two rings, however, are plainly distinguishable, e.g., by their energy patterns or magnetization distributions.

Yet another type of ring structure is possible if the central meron has positive polarity (Figure 10). In this case, no CCW and/or CW varieties are formed, and the distribution of anti-merons with opposite vorticities around the circle is perfectly symmetric. All rings energetically benefit for a large number *N* of constituent bimerons. We dub such rings “roundabouts”, with the negative and the positive polarity of the central meron (i.e., “(−)roundabout” and “(+)roundabout”) used to distinguish them from a bimeron ring, in which all dipole moments point to the ring center—“crossing” (Figure 11).

Figure 8a, Figure 9a, Figure 10a and Figure 11a show the color plot for the $m_{z}$-component of the magnetization on the xy plane. Figure 8b, Figure 9b, Figure 10b and Figure 11b show the energy density of the DMI zoomed in some range to make the subtleties of the internal structure discernible. Figure 8c, Figure 9c, Figure 10c and Figure 11c show the total energy density distributions on the xy plane. Figure 8d, Figure 9d, Figure 10d and Figure 11d show the energy density of the easy-plane anisotropy (EPA). Figure 8e, Figure 9e, Figure 10e and Figure 11e show the exchange energy density as color plots on the xy plane.

Similar ring-shaped bimeron patterns were recently addressed in [61] through atomistic spin simulations on a twisted bilayer magnet CrCl_3_. It was highlighted that 2D vdW magnets with an additional twist between layers open a unique avenue for investigating the properties of bimerons with remarkable flexibility, either through external stimuli or through the creation of heterostructures.

Within the Skyrme model—a nonlinear field theory of pions which possesses topological solitons that describe baryons—it has been shown that a ringlike solution is formed to reduce the energy density peaks at the ends of the chains [62,63]. Interestingly, the transition between the two configurations occurs at the baryon number 15.

### 4.3. Ring-Shaped Bimeron Macromolecules—“Crossings”

The internal stability of a “crossing” hinges on the coupling between the exterior bimerons and the central meron, which in the case of N=6, depicted in Figure 11, should possess negative polarity. Such a state presumably has a counterpart when the circular bimerons face the interior but the central meron acquires the negative vorticity; however, this will be addressed elsewhere. The energetic disadvantage of a “crossing” arises due to the mutual repulsion among the exterior bimerons. Moreover, the coupling with the central meron fades with the increasing number of bimerons, as the minimum ΔΦ of the interaction potential (Figure 3a) becomes smaller. Such a ring favors a smaller number *N* of bimerons, contrary to the “roundabout” rings considered before. Certain parts with positive DMI energy due to the overlapped “wings” continue to remain in the “crossing” (the white dotted regions in Figure 11b).

### 4.4. Stability of Bimeron Macromolecules

All of the considered ring-shaped macromolecules are stable with respect to the symmetric scaling of the inter-bimeron distances, and all reach their energy minima. However, we can compute the total energy of a bimeron macromolecule only in the vicinity of the energy minimum; otherwise, the pinning of meron centers imposes structural deformations. Moreover, the interaction energy in the case of bimeron rings includes amounts due to the attracting inter-bimeron interaction and to the repulsion with the central meron. Hence, direct comparison with the interaction potential for a bimeron pair in Figure 5a is rather difficult.

Crossings exist in the range N∈[2,11] (the red line in Figure 12a), and the energy linearly grows with the increasing number of bimerons. For a large number *N*, the repulsion among exterior bimerons outweighs their coupling to the central meron, and crossings become unstable. As expected, the central meron with negative polarity bears the topological charge Q=−1/2 (the inset in Figure 12a).

The (−)roundabouts can be constructed starting from N=6 (the blue line in Figure 12a) and do not have any upper limit on their bimeron number. For N>8, the (−)roundabouts have lower energy than the crossings, as indicated by the intersection of the blue and red lines in Figure 12a. The central meron with negative polarity bears the topological charge Q=−1/2 (the inset in Figure 12a).

The (+)roundabouts are stable even for a smaller number of bimerons, e.g., for N=5 (the green line in Figure 12a), and are energetically more favorable in the whole range of *N* compared with their negative counterparts. The central meron with positive polarity bears the topological charge Q=+1/2 (the inset in Figure 12a).

The large angle between bimeron dipoles prevents the formation of rings with small bimeron numbers *N*. Chains have the lowest energy among all bimeron macromolecules (the black line in Figure 12a), and do not have any restrictions on the smallest and/or largest number of constituent solitons.

Figure 12a shows the total energies and topological charges of bimeron macromolecules, excluding the edge states formed at the sample boundaries. Corresponding values for the included edge states are shown in Figure 12b. First of all, we note that the edge states add some positive amount to the topological charges: Q=2/3 and Q=−1/3 (the blue and green lines in the inset). Then, despite being far away from the bimeron configurations, the edge states favor (+)roundabouts over chains, a trend which would become even more pronounced for samples with cylindrical geometry. In small samples, the chains would come into contact with the edge states and presumably swirl into symmetric round structures; however, this phenomenon will be addressed elsewhere.

The considered bimeron macromolecules may become unstable if a central meron is permitted to shift from its balanced position in the ring center coupled to the surrounding particles. This unstable mode can tear ring-shaped clusters apart in an attempt to stabilize the more stable chain-macromolecules. Depending on the polarity of the central meron, it either beats any two exterior bimerons out of their shallow energy minimum, defined by ΔΦ, or merges with one of the exterior merons with the opposite vorticity (see Appendix A).

For example, the Q=+1/2 central meron in Figure 12c annihilates by merging with a (Q=−1/2)-anti-meron (both merons have the same polarity and appear red in the color plots of the magnetization). Thus, the final topological charge does not change, and equals Q=−4.5 for N=5 (Figure 12d). Interestingly, the central meron in this case performs a circular movement around the ring center with increasing amplitude until it breaks the ring into a chain (see Appendix A).

The total topological charge remains intact for N=7 exterior bimerons within the (−)roundabout (Figure 12e), and amounts to Q=−7.5 (from eight circular anti-merons and seven anti-merons with negative vorticity; see Figure 12f). In this case, no bimeron annihilation occurs, as all merons possess the same topological charge of −1/2. The final state represents a deformed chain (see Appendix A).

The crossing with N=5 exterior bimerons (Figure 12g) rearranges into a macromolecule which consists of a crossing with N=3 and parts of chains attached to it (Figure 12h, and see Appendix A).

The considered transformations of bimeron macromolecules provide information on the initial and final states that can be used to calculate the energy barrier with the geodesic nudged elastic band method [64]; however, this will be done elsewhere.

The considered transformations comply well with the stability arguments for ring-shaped macromolecules highlighted before: (i) the crossings prefer smaller numbers of bimerons; (ii) on the contrary, the roundabouts improve their stability for larger *N*. In Figure 12a,b, the solid lines indicate macromolecules which are robust in terms of both the distance scaling among bimerons and the instability mode of the central meron. The crossings are internally stable for N=3,4, the (−)roundabouts for N≥8, and the (+)roundabouts for N≥6. For N=5, only chains are possible.

## 5. Bimeron “Polymers”

In this section, we consider numerous bimerons filling a space with some density and reflecting the principles drawn in the previous sections. First, we attempt to obtain ordered bimeron polymers.

### 5.1. Combination of “Roundabouts” and “Crossings”

Bimeron macromolecules can be designed through a combination of “roundabouts” and “crossings”, which introduces frustration into the system. Indeed, the equilibrium inter-bimeron angle in “crossings” is defined as 2π/N, whereas in roundabouts it is π−2π/N. In the simplest case, both arrangements require the same angle between the constituent bimerons.

Figure 13a shows a “benzene” macromolecule: the (+)roundabout (highlighted by the dotted red circle) with the angle 2π/3 between bimeron dipoles additionally attracts six bimerons to form “crossings” (one of the crossings is highlighted by the tripod with solid red lines) with the same mutual angle. The formed macromolecule is highly symmetric and resembles the structure of benzene: C_6_H_6_. For only three bimerons attached to the (+)roundabout (Figure 13b), the resulting macromolecule deforms, i.e., the mutual inter-bimeron angle varies around the ring, with interstitial bimerons being slightly drawn into the interior of the ring (the ideal tripod is shown by the black solid lines). For other combinations of crossings around the (+)roundabout, the macromolecules become unstable and break apart (see Appendix A).

Interestingly, a symmetric macromolecule can be obtained for N=8 (Figure 13c). In this case, the mutual angle among the bimerons within the ring equals 3π/4, as would be the case in a (+)roundabout without additional bimerons attached. For only four bimerons attracted to the (+)roundabout (Figure 13d), however, the situation is similar to Figure 13b; the centers of some bimerons lay on the circle (dotted red line), whereas the centers of other bimerons are within this ring. Macromolecules with smaller numbers of crossings (such as 2, Figure 13e) exhibit internal stability, although accompanied by transformation of the ring into an ellipse. In addition, it is possible to speculate about one-dimensional chain-like structures composed of bimeron rings (Figure 13f).

Already, these simple examples show that only a restricted number of bimeron macromolecules can be constructed; otherwise, the system disintegrates into simpler random motives of bimerons.

As the next instructive example, we consider the bonding process of bimeron chains (Figure 14a). As an initial state, we use three parallel chains with an equal number of bimerons which are linked by two crosspieces. Although the resulting bimeron macromolecule deforms after the relaxation process and exhibits the frustration of inter-bimeron angles described before, the interior part of the molecule features (N=4) “crossings” (highlighted by the red cross) with the angle π/2 between the bimerons. This opens up the possibility of constructing a larger bimeron cluster, the interior of which would exhibit a perfectly square arrangement of bimerons (Figure 14b), whereas the boundary of such a cluster would remain distorted. A hexagonal arrangement of bimerons would be possible as well (Figure 14c). We note, however, that in these cases at least two bimerons are needed to form each side of the square or hexagonal unit cell.

### 5.2. Periodic Tessellations

We found it impossible to tessellate the whole space with roundabouts, which contain only one bimeron along the cell side. Figure 15 (first panel) shows a hexagonal arrangement of (+)roundabouts and (−)roundabouts which would be a candidate for a periodic bimeron polymer. Such a bimeron order, however, transforms into a disordered state (Figure 15, and see Appendix A).

Thus, in order to make periodic bimeron structures one must “dilute” the number of “crossings” and make the sides of the hexagonal or square cells longer, as in Figure 14b,c. We note that the periodicity is defined by the equilibrium inter-bimeron distance, and would not exhibit any energy minimum with the increasing period of the lattice. On the contrary, bimerons formed within the mentioned Skyrme model [62,63] form a square lattice of half-charge lumps, which constitutes the minimal-energy crystal structure for the easy-plane potential. In the same way, a meron cluster with a square lattice of vortices and antivortices minimizes the energy for frustrated magnets with easy-plane anisotropy [35].

### 5.3. Interconnection between Bimeron Polymers and the Hexagonal Skyrmion Lattice

Finally, we address the mutual transformation between the hexagonal Skyrmion lattice and a bimeron polymer.

First of all, in Figure 16a–d, we reproduce the solutions for the SkL from [65], which, in the yellow-shaded region of the phase diagram (Figure 2a), represent the global minima of the functional (Equation 7). To make the processes occurring in the inter-Skyrmion regions visible, we plot the topological charge density $\rho_{Q}$. It can be seen that, as the easy-plane anisotropy increases, “seeds” of merons are formed with the opposite topological charge densities and opposite vorticities but the same positive polarity (the area within the black circle in (d) bears negative topological charge, whereas the dark red regions are positively charged).

Further, in our numerical simulations we increased the anisotropy value from −0.5 to −1.5, using the equilibrium SkL as an initial state. However, we argue that the same anisotropy variation can be achieved experimentally. For example, in layers of chiral liquid crystals [41], switching of the surrounding phase from the in-plane to the out-of-plane orientation has been demonstrated using ambient-intensity unstructured light. The bimeron chains were shown to transform into Skyrmion clusters. Mutual transformation between Skyrmions and bimerons has been observed in an applied magnetic field in a cubic helimagnet Fe_0.5_Co_0.5_Ge [40], where the shape anisotropy fulfilled the role of the easy-plane anisotropy. The uniaxial anisotropy can be tuned by applying perpendicular strain in van der Waals magnetoelectric heterostructures, and varies in a wide range from −3.5 to 1.6 meV, that is, from the easy-plane to the easy-axis characteristic [42]. The creation and annihilation of bimerons has been achieved via perpendicular strain and an electric field without an external magnetic field [42].

During the transformation process shown in Figure 16e–i, six merons and six anti-merons acquire their complete shape from the described nuclei within the inter-Skyrmion regions of the SkL; the merons are located in the corners of hexagonal cells, whereas the anti-merons are located in the regions between two Skyrmions (Figure 16f). Because they bear opposite topological charges, the formed merons and anti-merons collapse pairwise. First, the SkL becomes distorted (Figure 16g), enabling merons and anti-merons to draw nearer. In Figure 16h, some of the meron pairs oriented along the *x* axis have already collapsed (shown by white ellipses). Eventually, all such pairs collapse, leaving a balanced number of merons which appear from the domain walls and merons from the centers of the lattice cells (Figure 16i, Appendix A).

The reverse transformation (Figure 16j–n), i.e., for the uniaxial anisotropy switched from −1.5 back to −0.5, would occur in a different manner; circular merons with negative polarity would elongate and fill the whole 2D plane when placed into the region of the phase diagram with a stable cycloidal state (h=0) or into the SkL (h=0.5), while the anti-merons with positive polarity would squeeze back into the domain-wall regions. In general, the created Skyrmions are deformed (elongated) due to the small number of bimerons within polymers. Indeed, these Skyrmions may either form an ideal hexagonal lattice (though with a larger period as compared with the equilibrium one) or elongate to fill the inter-Skyrmion voids with one-dimensional twists. The latter scenario turns out to be energetically advantageous; nonetheless, a perfect SkL can be formed when the elongated Skyrmions cut and consequently employ the first type of bimerons [32] discussed in the introduction. For a sufficient number of bimerons, the created SkL demonstrates an almost perfect arrangement of hexagonal cells (Figure 16n, Appendix A).

## 6. Conclusions

In the present paper, we have examined the properties of bimerons and the mechanisms leading to their condensation into the extended clusters we have dubbed “polymers” in two-dimensional chiral magnets with easy-plane anisotropy.

First of all, we revisit the internal structure of bimerons and base our analysis mainly on energy profiles in the xy plane, which is a conventional approach to addressing inter-soliton potentials. The characteristic pattern of the energy density exhibits a circular region with reversed rotational sense of the magnetization, which is located behind the anti-vortex. This region cannot be understood merely as the domain boundary formed with respect to the homogeneous in-plane state; rather, it represents a characteristic feature of the magnetization rotation which, starting from the anti-vortex center, continues to rotate below the $\left(m_{z}=0\right)$-level before returning with the reverse twist.

Second, we show that metastable bimerons, driven by their tendency to reduce the positive energy amount within circular regions with the “wrong” rotational sense, develop attracting interaction and naturally gather into chains. Then, the wrong twist area of each bimeron is covered by its preceding neighbor, which reduces the total energy of such bimeron “macromolecules”. Because one circular region always retains by the leading bimeron within a chain, bimerons may alternatively swirl into ring-shaped macromolecules. The concept of rings is not new, and appears, for example, within the Skyrme model for baryons. In the present case of easy-plane bimerons, however, we argue that there are several varieties of such ringlike solutions, which we classify and name according to the fashion in which the bimeron dipoles align around the center. In (±)“roundabouts”, the bimeron dipoles are oriented along the circle; they may swirl clockwise and counterclockwise, which is energy-degenerate. In our terminology, (+) and (−) point to the polarity of the meron inevitably formed in the center of such a roundabout. In “crossings”, bimeron dipoles point into the ring-center. We have scrutinized the internal structure and stability of different types of such rings depending on their number of constituent bimerons, finding that although rings may be stable with respect to their minimization of the inter-bimeron distances and constitute local energy minima, they may be destroyed by other instability modes. For example, a (+)roundabout with a small number of bimerons *N* prefers to merge its central meron with one of the anti-merons within the boundary, and consequently transforms into a chain. The central anti-meron in a (−)roundabout may simply rupture the boundary and lead to a chain solution. “Crossings” with a large number *N* of constituent bimerons easily turn into “crossings” with a smaller number connected by patches of chains. As a result, the family of “crossings” is represented only by (N=3) and (N=4) members, whereas roundabouts gain their stability only starting from (N=8).

In addition, we studied bimeron macromolecules characterized by combinations of crossings and roundabouts, which introduces some frustration of inter-meron angles. Thus, such macromolecules are usually deformed, and cannot tessellate the whole space. However, periodic bimeron lattices are possible as balanced tessellations of “crossings”, chains, and “roundabouts”. We achieved hexagonal and square bimeron lattices when the sides of unit cells were formed by several bimerons, i.e., N>1; otherwise, such lattices are torn into disordered polymers.

Moreover, we identified the exact scenario of inter-transformation between hexagonal SkLs and disordered bimeron polymers. In the direct transformation from the SkL, pairs of merons and anti-merons are nucleated within the cell boundaries. As they have the opposite topological charges, which facilitates nucleation, these merons merge and disappear pairwise, leaving a disordered state of bimerons. In the reverse process, circular bimerons with negative polarity become energetically favorable, and are consequently redistributed within the plane, whereas antivortices squeeze into the boundaries of the formed Skyrmion lattice.

We argue that our findings can shed light on the well-studied phase diagram for quasi-two-dimensional chiral magnets with easy-plane anisotropy, and can complement previous studies from both fundamental and applied points of view.

## Figures and Tables

**Figure 1 nanomaterials-14-00504-f001:**
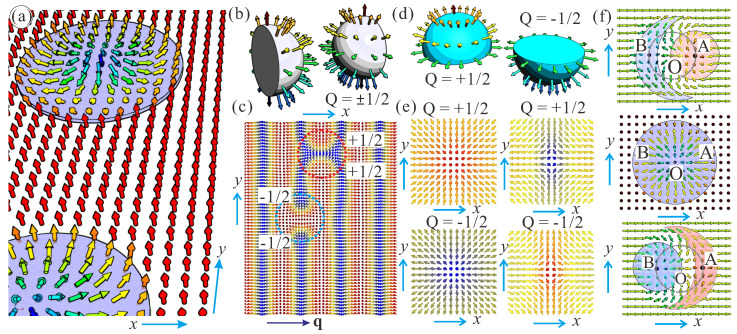
(**a**) Schematics of isolated Néel Skyrmions in polar magnets with $C_{nv}$ symmetry (or in multilayers with the induced DMI). (**b**,**c**) Schematics of bimerons formed as ruptures of the cycloidal spiral according to (Equation 3). The magnetization field wraps only the corresponding half of the S_2_ sphere (**b**). The upper anti-meron within the blue circle in (**c**) and the lower meron within the red circle can mutually annihilate, leaving a bimeron pair with total charge Q=0. (**d**,**e**) Schematics of bimerons formed as a result of wrapping the upper or lower hemisphere. There are two varieties, merons and anti-merons, with positive and negative vorticity m=±1. (**f**) Schematics of two bimerons with opposite polarity obtained from the axisymmetric Skyrmion (middle panel) by magnetization rotation around the *y* axis with the angle π/2. The magnetization in the center of the isolated Skyrmion (the middle panel in (**f**), point *O*) now points horizontally along the dipole moment of the bimeron in the upper and lower panels of (**f**). Two points *A* and *B* with $m_{z}=0$ within the isolated Skyrmions (the middle panel in (**f**)) now become the centers of circular and crescent-shaped merons (the upper and lower panels of (**f**)).

**Figure 2 nanomaterials-14-00504-f002:**
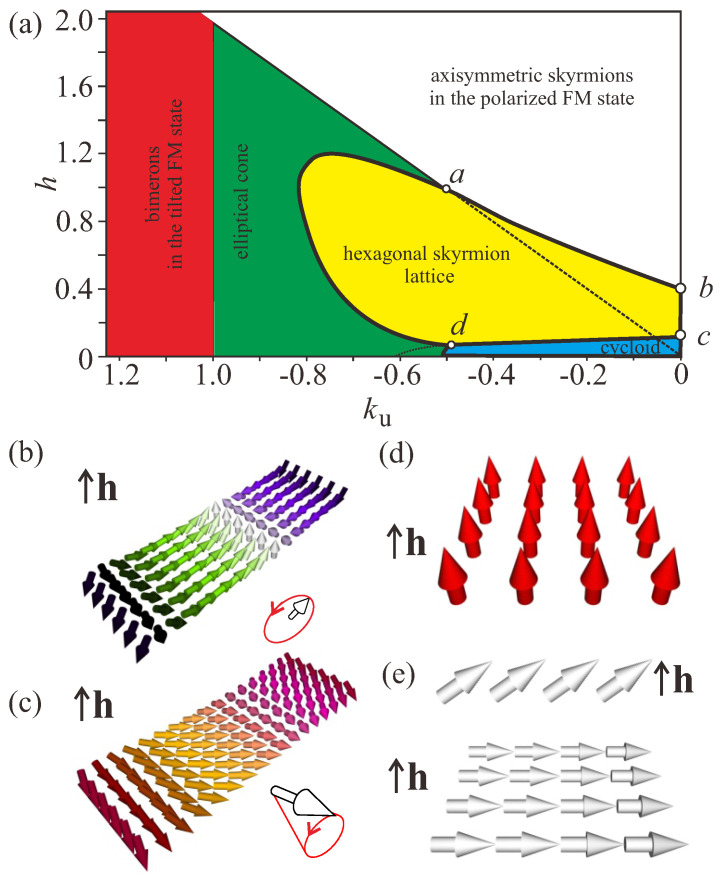
(**a**) Magnetic phase diagram of the solutions for model (Equation 7) with easy-plane uniaxial anisotropy (EPA). The filled areas designate regions of thermodynamic stability of the corresponding phases: blue shading—cycloidal spiral (**b**); green shading—elliptical cone (**c**); white shading—polarized ferromagnetic state (**d**); yellow shading—hexagonal Skyrmion lattice; red shading—tilted ferromagnetic state (**e**). The thick black lines indicate the first-order phase transitions between corresponding phases, while the thin black lines indicate the second-order phase transitions. The field is measured in units of $H_{0}=D^{2}/A\left|M\right|$, i.e., $h=H/H_{0}$, and $k_{u}=K_{u}M^{2}A/D^{2}$ is the non-dimensional anisotropy constant.

**Figure 3 nanomaterials-14-00504-f003:**
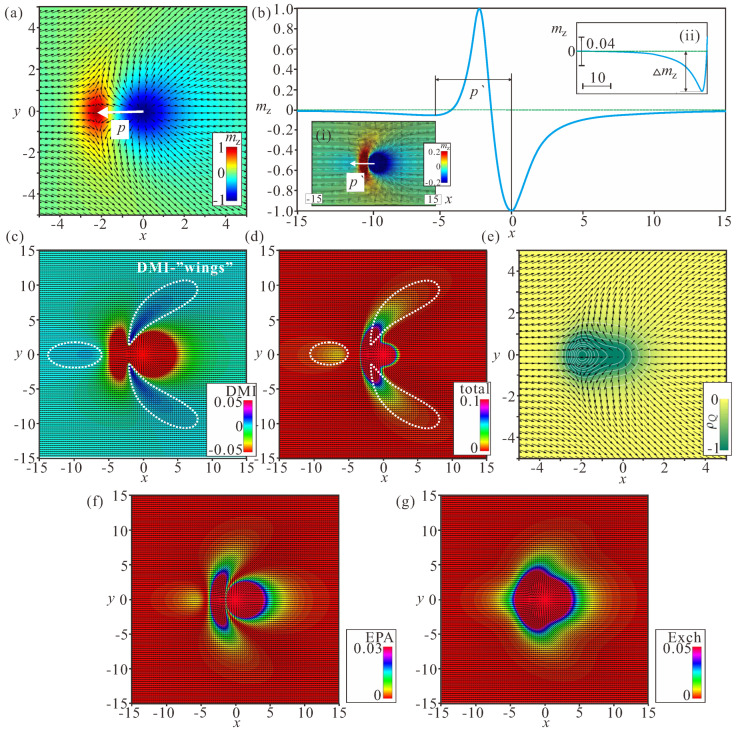
Internal magnetic structure of an isolated bimeron characterized by the xy-color plots of the $m_{z}$-component of the magnetization (**a**), the DMI and total energy densities (**c**,**d**), and the topological charge density (**e**); h=0, $k_{u}=-1.5$. The cross-cut (**b**) along the meron center shows the part with the opposite rotational sense, the center of which is located at distance $p^{\prime }$ from the center of the circular meron, with *p* being the distance between the centers of the vortex and antivortex. The insets in (**b**) show corresponding zoomed 2D (**i**) and 1D (**ii**) magnetization distributions, i.e., in (**i**) the color indicates the magnetization value in the range [−0.2;0.2]. The corresponding color plots for the exchange energy density and the anisotropy energy densities are shown in (**f**,**g**). The Zeeman energy is 0, as h=0, while the field is measured in the units of $H_{0}=D^{2}/A\left|M\right|$, i.e., $h=H/H_{0}$. The magnetization vector m(x,y)=M/|M| is normalized to unity; $k_{u}=K_{u}M^{2}A/D^{2}$ is the nondimensional anisotropy constant; the spatial coordinates x are measured in units of the characteristic length of modulated states $L_{D}=A/D$; and the value $\lambda =4\pi L_{D}$ for zero magnetic field is the period of the cycloid.

**Figure 4 nanomaterials-14-00504-f004:**
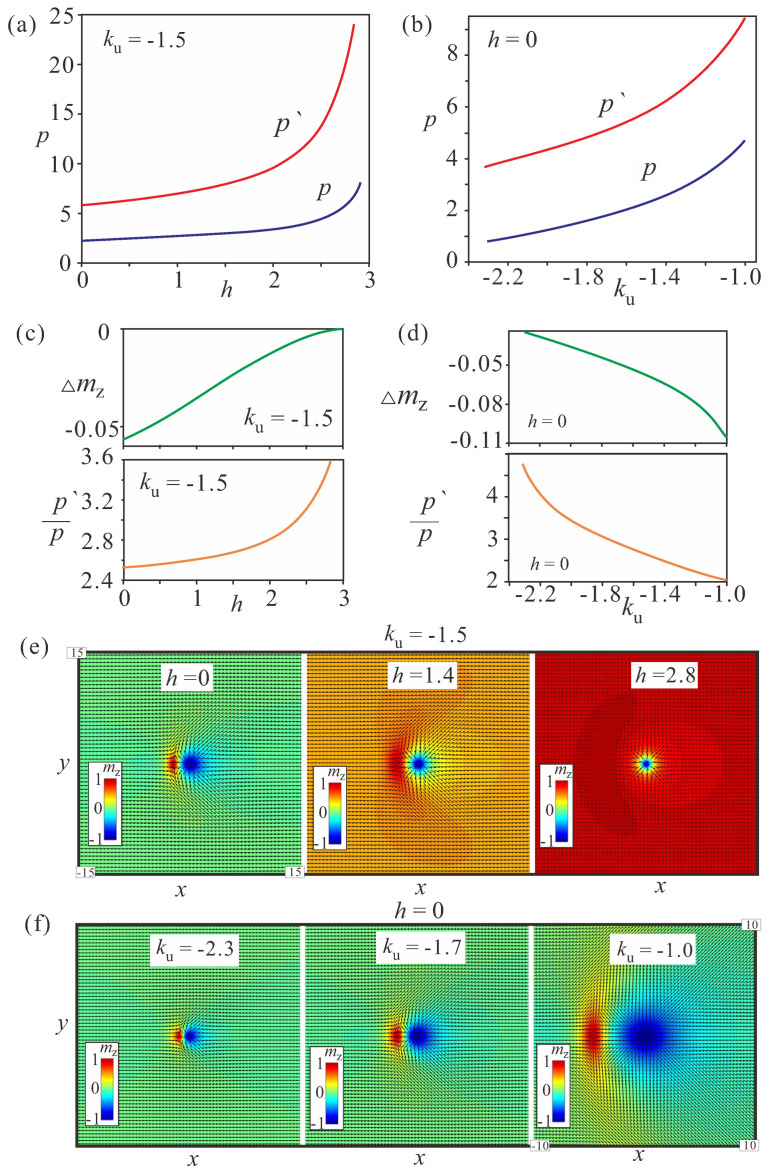
(**a**,**b**) Field- and anisotropy-driven evolution of the parameters *p* and $p^{\prime }$ for a fixed value of the anisotropy ($k_{u}=-1.5$, (**a**)) and field (h=0, (**b**)), correspondingly; *p* is the distance between the centers of the vortex and the antivortex, while $p^{\prime }$ is the distance from the center of the circular meron to the center of the circular area with the wrong rotational sense. For the field-driven transformation of bimerons, the parameters *p* and $p^{\prime }$ as well as their ratio (**c**) all increase. This means that the parameter $p^{\prime }$ increases faster than the parameter *p*, with both processes indicating the transformation into an isolated Skyrmion surrounded by a homogeneous state with $m_{z}=1$. The depth of the region with the wrong twist $\Delta m_{z}$ tends to zero (**c**). Qualitatively, the same process occurs with growing anisotropy (**b**,**d**). At some critical anisotropy value, the bimerons become too small to be addressed using the chosen cell sizes of the numerical grids. The color plots of the magnetization in (**e**,**f**) reflect the above-mentioned field- and anisotropy-driven transformations. The black arrows in all color plots are the projections of the magnetization onto the xy plane.

**Figure 5 nanomaterials-14-00504-f005:**
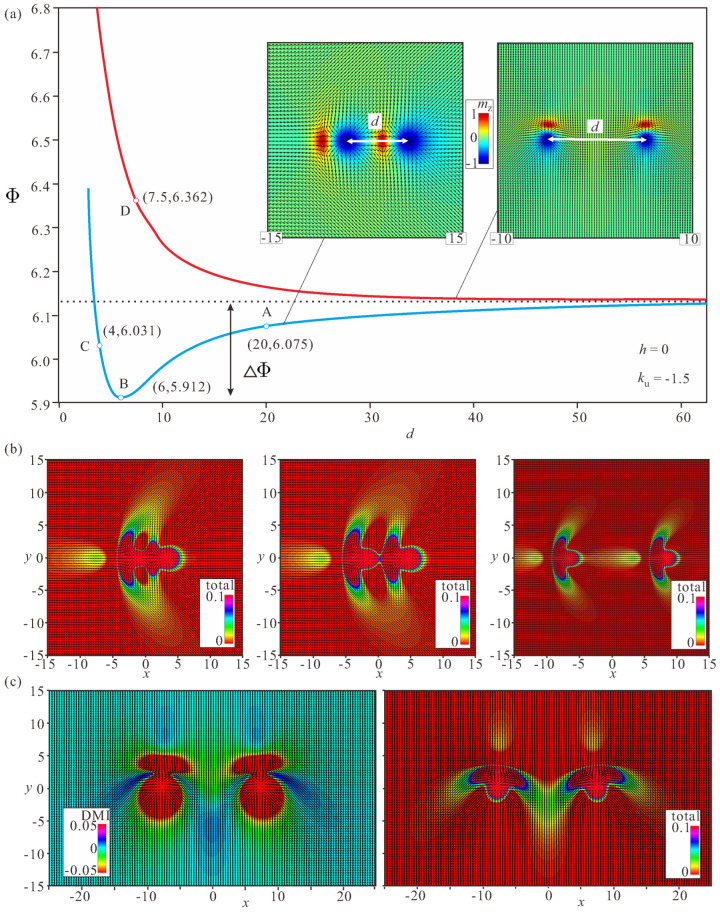
(**a**) The inter-bimeron potential Φ versus the distance *d* between the centers of circular anti-merons for two bimerons oriented head-to-tail (blue curve and left inset) and/or side-by-side (red curve and right inset). The color plots for the total energy density (**b**) are shown for bimerons located at larger distances (point *A*, right panel), in the minimum (point *B*, middle panel), and at shorter distances (point *C*, left panel). These color plots indicate the underlying reason for the attracting interaction and the minimum of Φ; at some optimal inter-meron distance, the first bimeron covers the circular region with the positive energy density of the subsequent bimeron. The color plots of the total energy and DMI energy density (**c**) do not demonstrate any energy benefit from coupling two bimerons. On the contrary, overlapping “wings” lead to an energy increase (red curve in (**a**)). Here, $h=0,k_{u}=-1.5$.

**Figure 6 nanomaterials-14-00504-f006:**
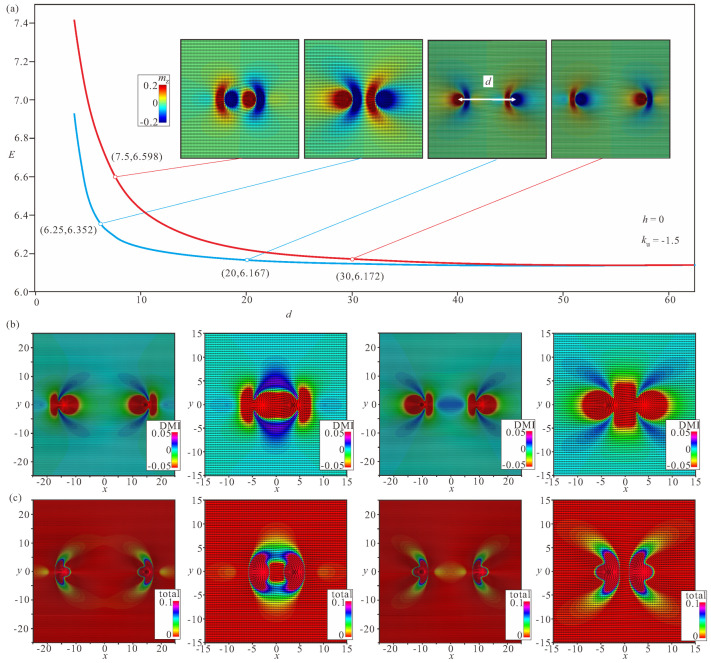
(**a**) The interaction potential for two bimerons with the opposite polarities exhibits only the inter-particle repulsion. Here, we orient two bimerons with either their circular merons (the red curve and left panels in (**b**,**c**)) or with their crescents (the blue curve and right panels in (**b**,**c**)) facing the inter-meron area. The insets show color plots of the magnetization at the indicated points of the curves. To plot both curves, we pinned the magnetization in the centers of the circular anti-merons. Obviously, being unpinned, the two bimerons find a path to annihilate, as they have the opposite topological charges. The corresponding DMI and full energy density distributions are plotted in (**c**). h=0, $k_{u}=-1.5$.

**Figure 7 nanomaterials-14-00504-f007:**
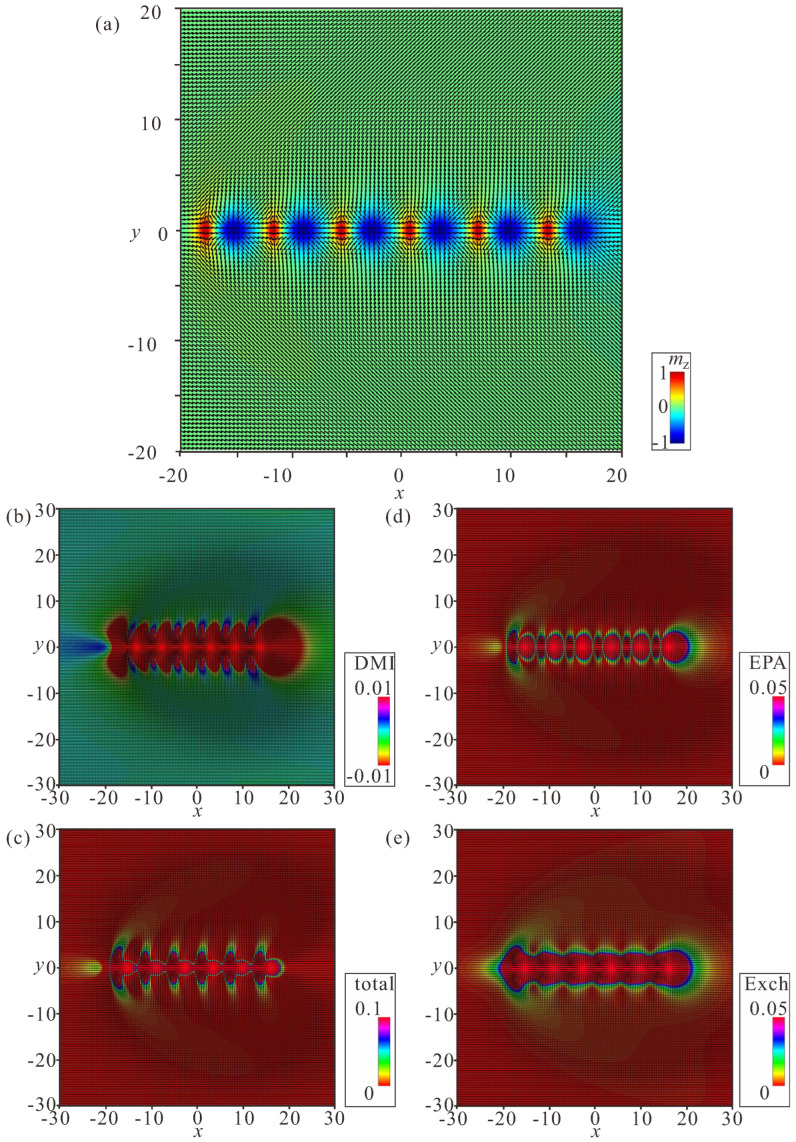
Internal structure of a bimeron chain with six constituent bimerons. (**a**) Color plot for the $m_{z}$-component of the magnetization. (**b**) Color plot for the DMI energy density. (**c**) Color plot for the total energy density. (**d**) Color plot for the energy density of the easy-plane anisotropy. (**e**) Color plot for the exchange energy density. h=0, $k_{u}=-1.5$. The field is measured in units of $H_{0}=D^{2}/A\left|M\right|$, i.e., $h=H/H_{0}$. The magnetization vector m(x,y)=M/|M| is normalized to unity. $k_{u}=K_{u}M^{2}A/D^{2}$ is the nondimensional anisotropy constant. Spatial coordinates x are measured in units of the characteristic length of modulated states $L_{D}=A/D$. The value $\lambda =4\pi L_{D}$ for zero magnetic field is the period of the cycloid.

**Figure 8 nanomaterials-14-00504-f008:**
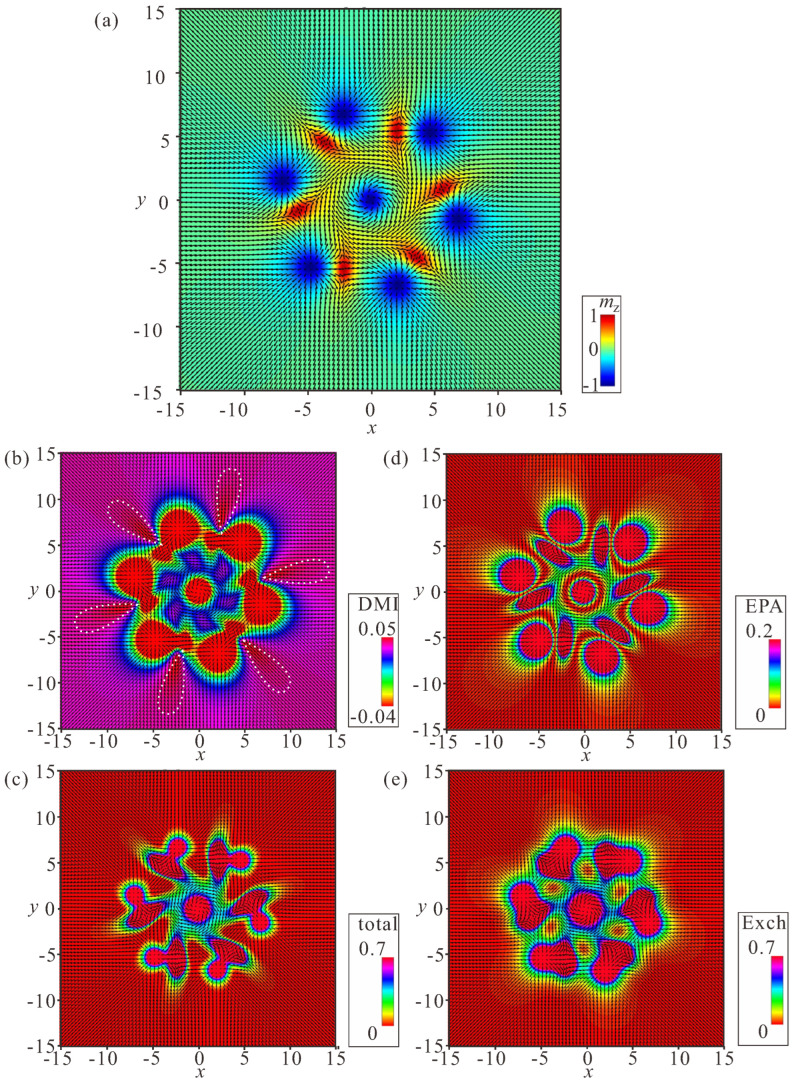
Internal structure of a (−)roundabout with counterclockwise circling of bimerons. (**a**) Color plot for the $m_{z}$-component of the magnetization. (**b**) Color plot for the DMI energy density. (**c**) Color plot for the total energy density. (**d**) Color plot for the energy density of the easy-plane anisotropy. (**e**) Color plot for the exchange energy density. h=0, $k_{u}=-1.5$. The field is measured in units of $H_{0}=D^{2}/A\left|M\right|$, i.e., $h=H/H_{0}$. The magnetization vector m(x,y)=M/|M| is normalized to unity. $k_{u}=K_{u}M^{2}A/D^{2}$ is the nondimensional anisotropy constant. Spatial coordinates x are measured in units of the characteristic length of modulated states $L_{D}=A/D$. The value $\lambda =4\pi L_{D}$ for zero magnetic field is the period of the cycloid.

**Figure 9 nanomaterials-14-00504-f009:**
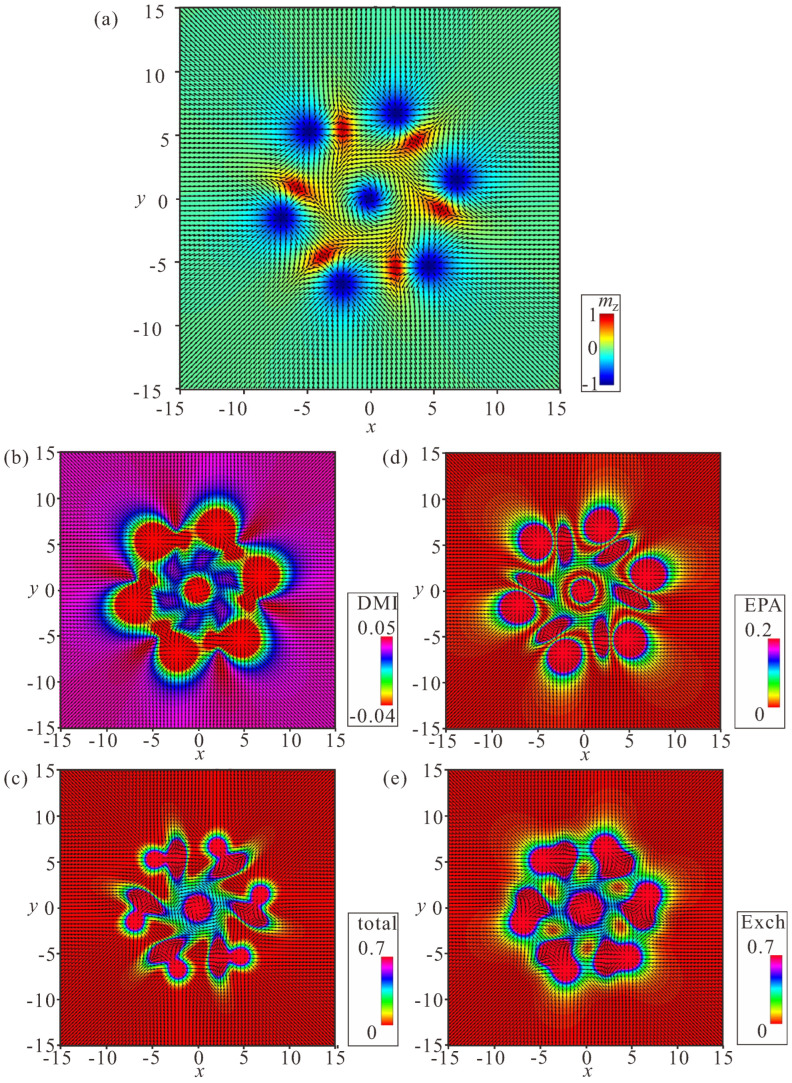
Internal structure of a (−)roundabout with clockwise circling of bimerons. (**a**) Color plot for the $m_{z}$-component of the magnetization. (**b**) Color plot for the DMI energy density. (**c**) Color plot for the total energy density. (**d**) Color plot for the energy density of the easy-plane anisotropy. (**e**) Color plot for the exchange energy density. h=0, $k_{u}=-1.5$. The field is measured in units of $H_{0}=D^{2}/A\left|M\right|$, i.e., $h=H/H_{0}$. The magnetization vector m(x,y)=M/|M| is normalized to unity. $k_{u}=K_{u}M^{2}A/D^{2}$ is the nondimensional anisotropy constant. Spatial coordinates x are measured in units of the characteristic length of modulated states $L_{D}=A/D$. The value $\lambda =4\pi L_{D}$ for zero magnetic field is the period of the cycloid.

**Figure 10 nanomaterials-14-00504-f010:**
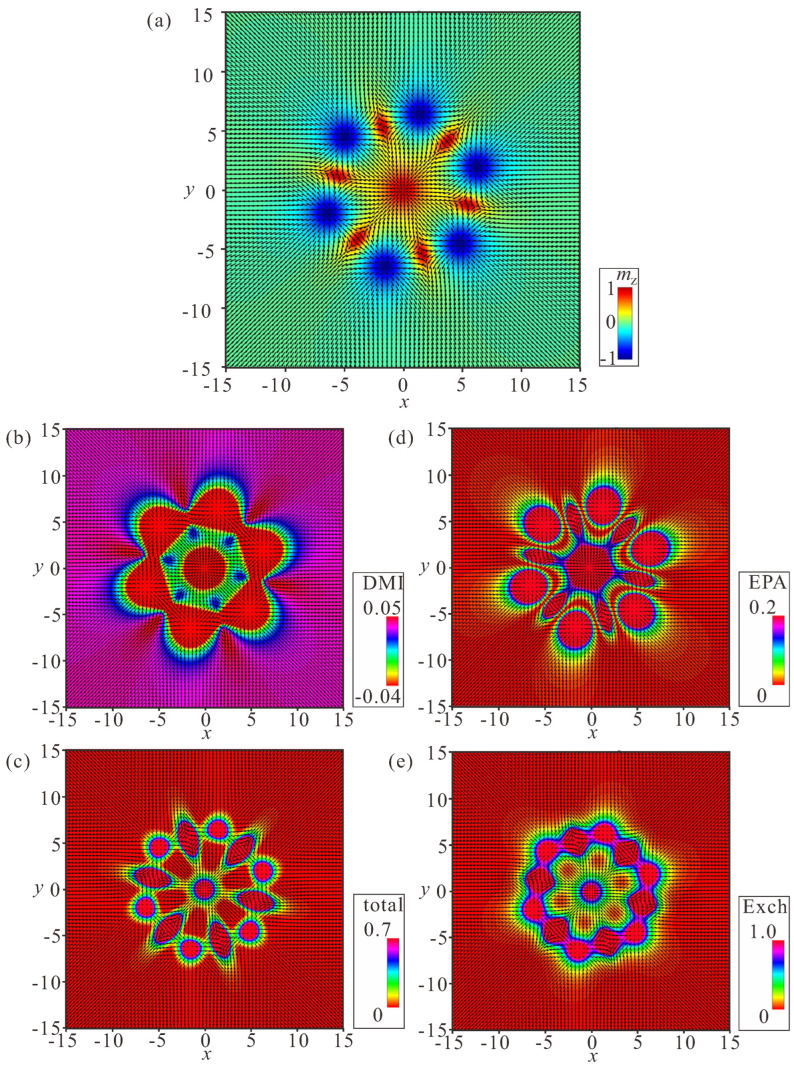
Internal structure of a (+)roundabout. (**a**) Color plot for the $m_{z}$-component of the magnetization. (**b**) Color plot for the DMI energy density. (**c**) Color plot for the total energy density. (**d**) Color plot for the energy density of the easy-plane anisotropy. (**e**) Color plot for the exchange energy density. h=0, $k_{u}=-1.5$. The field is measured in units of $H_{0}=D^{2}/A\left|M\right|$, i.e., $h=H/H_{0}$. The magnetization vector m(x,y)=M/|M| is normalized to unity. $k_{u}=K_{u}M^{2}A/D^{2}$ is the nondimensional anisotropy constant. Spatial coordinates x are measured in units of the characteristic length of modulated states $L_{D}=A/D$. The value $\lambda =4\pi L_{D}$ for zero magnetic field is the period of the cycloid.

**Figure 11 nanomaterials-14-00504-f011:**
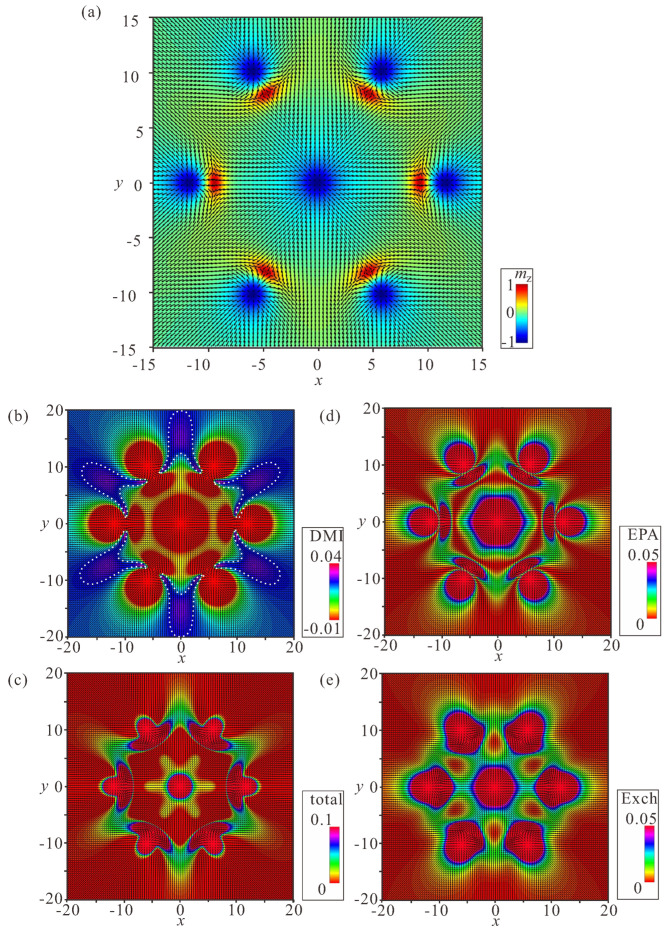
Internal structure of a bimeron “crossing”. (**a**) Color plot for the $m_{z}$-component of the magnetization. (**b**) Color plot for the DMI energy density. (**c**) Color plot for the total energy density. (**d**) Color plot for the energy density of the easy-plane anisotropy. (**e**) Color plot for the exchange energy density. h=0, $k_{u}=-1.5$. The field is measured in units of $H_{0}=D^{2}/A\left|M\right|$, i.e., $h=H/H_{0}$. The magnetization vector m(x,y)=M/|M| is normalized to unity. $k_{u}=K_{u}M^{2}A/D^{2}$ is the nondimensional anisotropy constant. Spatial coordinates x are measured in units of the characteristic length of modulated states $L_{D}=A/D$. The value $\lambda =4\pi L_{D}$ for zero magnetic field is the period of the cycloid.

**Figure 12 nanomaterials-14-00504-f012:**
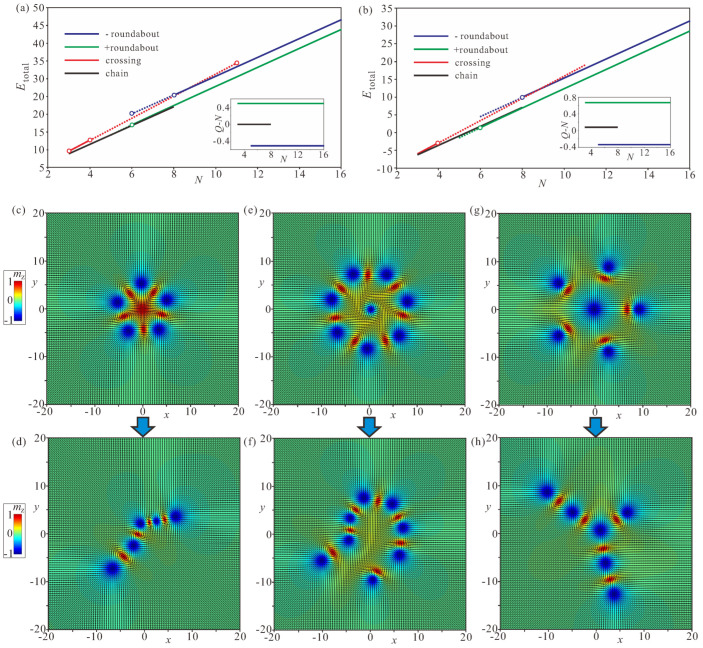
(**a**) Total energy of different bimeron macromolecules in dependence on the number *N* of constituent bimerons. The energy of edge states formed at the specimen boundary is excluded. The inset shows the topological charge of the central meron formed in circular macromolecules, computed as Q−N, where *Q* is the total charge of the magnetization distribution excluding the edge states and *N* is the number of exterior bimerons. The color coding is the same as used in the main graph: red for crossings, blue for (−)roundabouts, green for (+)roundabouts, and black for chains. For crossings and (−)roundabouts, the charges of the central merons are the same, Q=−1/2. Solid lines connecting points with a fixed number *N* indicate macromolecules which are robust against transformation into chains. Other macromolecules, indicated by dotted lines, can be wrapped into chains by the displacement of the central meron, i.e., such macromolecules are stable for symmetrical scaling of inter-meron distances but loose their stability while being deformed. (**b**) Total energy of bimeron macromolecules taking into account the energy of the edge states, which clearly favor (+)roundabouts. The edge states bear their own topological charges Q=0.16, which uniformly shift the dependencies in the inset. (**c**,**d**) Transformation of a (+)roundabout with N=5 into a buckled chain (see Appendix A). (**e**,**f**) Transformation of a (−)roundabout with N=7 into a looped chain (see Appendix A). (**g**,**h**) Transformation of a crossing with N=5 into a crossing with N=3 and fragments of attached chains (see Appendix A). $h=0,k_{u}=-1.5$.

**Figure 13 nanomaterials-14-00504-f013:**
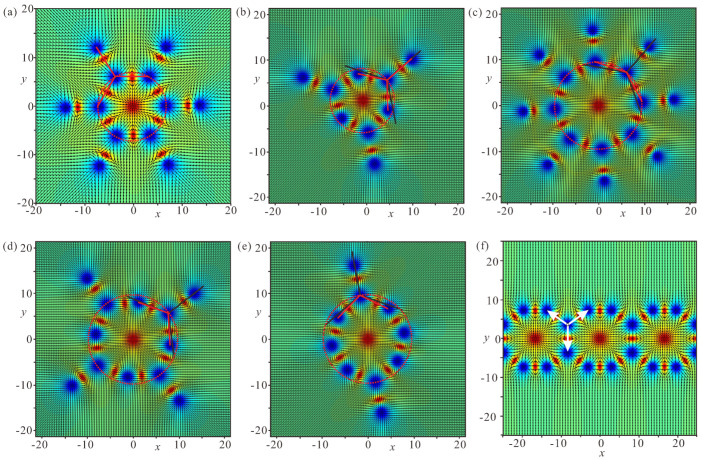
Stable bimeron macromolecules obtained by combinations of “crossings” and “roundabouts” for N=6 (**a**,**b**) and N=8 (**c**–**e**) in (+)roundabouts. In “benzene” (**a**), all angles between bimerons are 2π/3, as dictated by both ring varieties. In its (N=8) counterpart (**c**), however, the angle between the bimerons within the ring is 3π/4, as specified by the “roundabout”. Less symmetric macromolecules (**b**,**d**,**e**) exhibit structural deformations and the inter-bimeron angles vary around the rings, which may lead to instability (see Appendix A). In addition, (+)“roundabouts” with eight bimerons can alternatively be connected into a stripe—an analogue of the chain (**f**). $h=0,k_{u}=-1.5$.

**Figure 14 nanomaterials-14-00504-f014:**
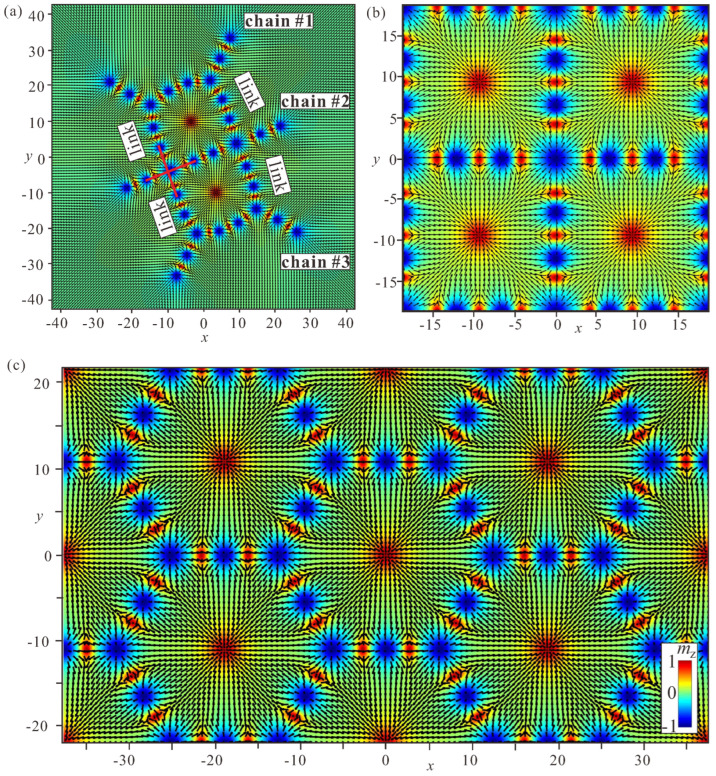
(**a**) The linking process of three chains with the same number of bimerons. The red cross highlights the (N=4) crossing with the inter-bimeron angles π/2, which facilitates the creation of a bimeron cluster with a square arrangement of bimerons (**b**). (**c**) Stable two-dimensional periodic tessellation with hexagonal ordering of bimerons. Such a state is possible only if the sides of the hexagonal cells contain several bimerons (two bimerons in the present case). $h=0,k_{u}=-1.5$.

**Figure 15 nanomaterials-14-00504-f015:**
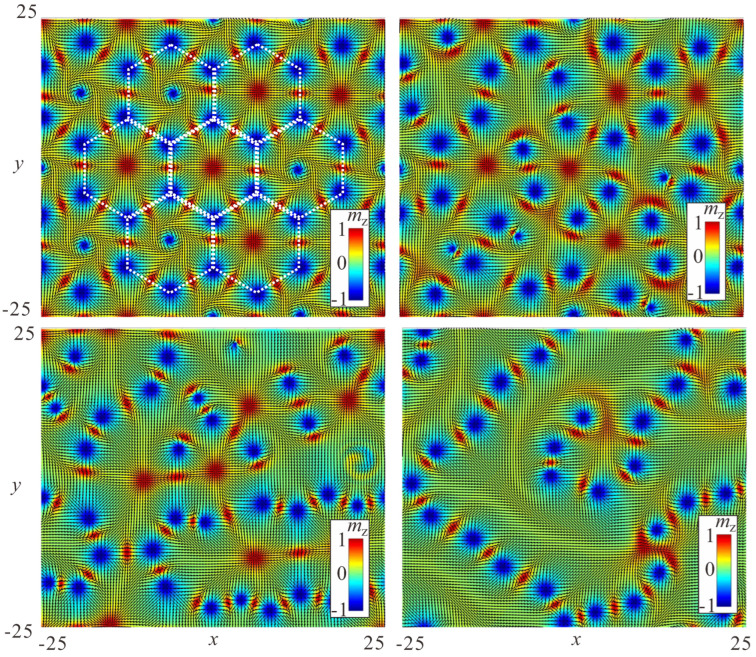
If the cell side of the periodic tessellations in Figure 14c contains just one bimeron, such a hexagonal order transforms into a random bimeron distribution. As an initial state in the first panel, we use a hexagonal bimeron lattice with a periodic mixture of (+)roundabouts and (−)roundabouts. Therefore, periodic boundary conditions can be used at both sides of the numerical grid. The subsequent panels demonstrate the disintegration process toward a random bimeron polymer (see Appendix A). $h=0,k_{u}=-1.5$.

**Figure 16 nanomaterials-14-00504-f016:**
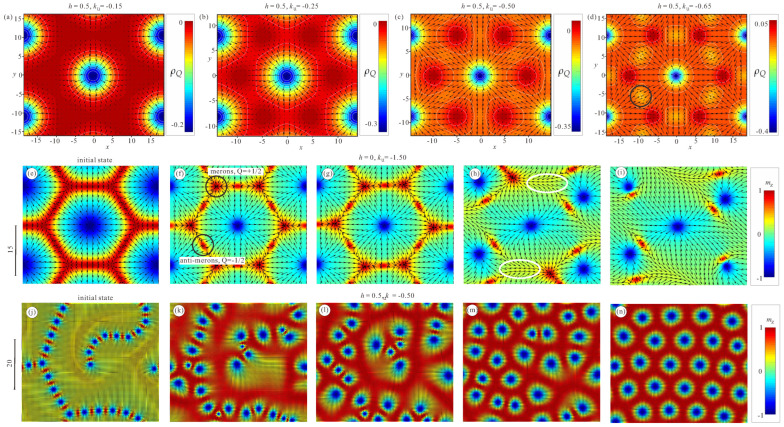
(**a**–**d**) The internal structure of the hexagonal SkL, shown as color plots of the topological charge density within the yellow-shaded region of the phase diagram in Figure 2a. With increasing easy-plane anisotropy (from left to right in the first row), the nuclei of merons with the opposite topological charges emerge within the cell boundaries. Mutual transformation between the hexagonal SkL and a disordered bimeron polymer is achieved by changing the uniaxial anisotropy from −0.5 to −1.5 (second row) and back from −1.5 to −0.5 (third row). The ordered SkL (**e**) is a local energy minimum for $k_{u}=-0.5,\;h=0$. If the anisotropy is suddenly switched to −1.5, then the SkL undergoes the following transformation: (**f**) merons and anti-merons with positive polarity nucleate pairwise within the cell boundary (highlighted by black circles); (**g**) the hexagonal cell becomes distorted, enabling merons and anti-merons to approach each other; (**h**) merons and anti-merons merge and annihilate (highlighted by white circles); and (**i**) only bimerons are left. The vortex originates from the Skyrmion in the center of the SkL, whereas the anti-vortex is a remainder of the boundary (see Appendix A). A disordered bimeron polymer (**j**) is a metastable cluster formed at $k_{u}=-1.5,\;h=0.5$. If the anisotropy switches to −0.5, then the SkL represents the global minimum of the energy functional (Equation 7). This means that the circular merons with polarity against the field become energetically favorable. These anti-merons rearrange and fill the whole space, whereas the anti-vortices with polarity along the field squeeze into the boundary regions (**k**,**l**). In (**m**), the Skyrmions form a disordered state but eventually manage to form an almost perfectly hexagonal arrangement (**n**) (see Appendix A).

## Data Availability

The data presented in this study are available on request from the corresponding author.

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
