# Peer review of "“Polymerization” of Bimerons in Quasi-Two-Dimensional Chiral Magnets with Easy-Plane Anisotropy"

_nanomaterials, 2024, doi:10.3390/nano14060504_

Round 1
Reviewer 1 Report
Comments and Suggestions for Authors
This work studies a chiral magnetic system with easy-plane anisotropy. This is known to support bimerons and the present paper gives makes some further significant observations and reports the existence of stable clusters of bimerons.
Sec. 3.2. explains the central and non-trivial finding that the energy for two bimerons has an energy minimum when the bimerons are at a specific distance from each other, i.e., they can form a bound pair. The importance of the finding is unfortunately obscured by many other trivial observations (explaining some other bimeron configurations that behave in a trivial way) that are extensively discussed in the same section.
Regarding the explanation for the stabilization of a bound bimeron pair, a lot of emphasis is placed in the "egg" regions that are plotted in many figures. There is though no solid argument why these explain the formation of bound bimeron pairs.
The argumentation about the explanation of the main finding is based on an observation made in Sec. 3.1 where an account of the "Internal Structure Bimerons" is given. The observation is that the magnetization component mz of a bimeron has an unexpected feature (turns from positive to negative before it decays to zero), as shown in Fig. 2b. Related is the observation that the swirling of the magnetization vector has some peculiarities close to the antivortex. I understand the argumentation and find it plausible. I am though not convinced that it is solid enough. It seems open that a different approach could as well prove that the above-mentioned feature only partially explains the numerical findings.
In any case, if the authors will insist in keeping this feature (generalized to what they call "egg" in later sections) in a central place in their work, they should probably compare this with a similar observation given in Ref. [41] where the bimeron configuration is described. (Are the corresponding features in the papers identical or only similar but not related?)
The discussion continues quite naturally and interestingly with numerical findings of "Bimeron macromolecules", in Sec. 4, and "Bimeron polymers" in Sec. 5. Sec. 4.1 discusses the very interesting case of stable chains of bimerons and it compares these with macromolucules. Sec. 4.2 and 4.3 discuss various complicated arrangements which seem interesting. Unfortunately, the descriptions of complicated arrangements of bimerons are largely heuristic. Also, there is extensive use of jargon like "eggs", "pilot", "co-pilot" and this makes the description and the arguments quite hard to follow.
The paper Lin et al, Phys. Rev. B 91, 224407 (2015) contains figures and some discussion regarding collections of bimerons. Some of the figures in that paper seem to show chains of bimerons. Do the authors think that these are related to their findings and to "macromolecules"?
I have the following questions (of varying importance).
Q1. Lines after Eq. (4). "one can also stabilize bimerons with Q = 0 by coupling merons and anti-merons." This is not substantiated. The authors should either support the statement or remove it.
Q2. page 4, first paragraph and Fig. 1f. Rotating the magnetization vector of a symmetric skyrmion (shown in Fig. 1f-center)should produce some type of symmetric bimeron configuration. It does not produce the crescents shown in Fig. 1f (as far as I can tell).
Q3. Sec. 3.1, 3rd line. What is meant here by "axisymmetric" (where does this feature refer to)?
Sec. 3.1, 3rd paragraph. What is "rotational energy"?
Q4. Sec 3.2, 3rd paragraph. We have merons and anti-merons, but, in the figure, we do not see "anti-merons towards each other" as stated in the text.
Q5. In Fig. 7, we have simulations with easy-plane anisotropy. It seems strange to see skyrmions here (which typically require easy-axis anisotropy). Is this some local minimum state, probably due to the specific boundary conditions?
What happens if we start the simulation from the state in Fig. 7f and switch the anisotropy back to -1.5?
Could you try switching the anisotropy to a value lower (in absolute value) than -0.5, e.g., to -0.3? Do you probably see something quite different in this case?
Q6. Sec. 6 "Conclusions", last paragraph: "our findings shed light on the well-studied phase diagram for quasi two-dimensional chiral magnets with the easy-axis anisotropy".
Is this a typo? Do you mean "easy-plane anisotropy"?
This also brings about the question about the phase diagram of an easy-plane chiral magnet, as in Eq. (7). This is, what are the minimum energy states as Ku varies? Discussing this issues seems like an appropriate small addition for this paper.
In conclusion, the important findings of this paper are that two merons can form a bound pair and, by implication, various clusters of merons form bound states. The description is detailed. Unfortunately, the language used makes some paragraphs hard to understand. (An example is on page 7 first paragraph, following Fig. 3. Another example is already in the first paragraph where we see the unclear terms "parental", "spotted" and later "outskirt".). I recommend this paper for publication in nanomaterials but I ask the authors to discuss and explain the issues mentioned above so that some obscure points are cleared and the paper becomes more complete.
Comments on the Quality of English Language
Some paragraphs are hard to understand. An example is on page 7 first paragraph, following Fig. 3. Another example is already in the first paragraph where we see the unclear terms "parental", "spotted" and later "outskirt".
There is extensive use of jargon like "eggs", "pilot", "co-pilot" and this makes the description and some arguments quite hard to follow.
Author Response
Kindly see the attached file

Reviewer 2 Report
Comments and Suggestions for Authors
The authors have studied interacting bimerons stabilized in a magnetic film with easy plane anisotropy and DMI interactions. This paper presents the results of micromagnetic calculations concerning: (i) the stability of a single bimeron as a function of the anisotropy constant and the value of the external field, (ii) the interaction of a pair of bimerons, and (iii) the formation of chain-like ("polymeric") systems of bimerons. The results presented are interesting and appear to be novel. The reviewer recommends publication of the revised manuscript in MDPI Nanomaterials. The authors should address the following comments:
1) Terminology:
The authors introduce a number of their own terms to describe the magnetic configuration and its energy landscape. The reviewer recommends the use of more descriptive and precise terms. The terms introduced by the authors should be removed from the abstract and the conclusion. For example, the phrases from the abstract "bimerons exhibit egg-shaped regions" or "bear the sense of rotation opposite to that chosen by the Dzyaloshinskii-Moriya interaction" are not clear.
2) Figures:
The paper contains complex drawings that are difficult to analyze:
-Fig. 1(f) - the y-axis around which the skyrmion texture is rotated is not included; regions A, B, O are not commented in the caption.
- Figures 2-7 - please mention in captions that coordinates, magnetization, field and anisotropy constant are expressed in units of A/D, M, D^2/(A M), D^2/(A M^2).
- Legends with colorbars are missing for some subfigures, units for the units in the legends are missing,
- Labels for subfigures (a), (b), (c), ..., and numbers are very small,
- The symbols p, p', Phi are not explained in the captions.
3) Model:
- Please provide a sketch of a phase diagram illustrating what phases of magnetic ordering are observable depending on the external field and the easy-plane anisotropy constant.
- It was not justified why dipole bending and dipole shape anisotropy were neglected.
- The direction in which the external magnetic field was applied was not given,
- no definition of the inter-bimeron potential was given; its relation to the energy density Eq. 7 was not explained; its unit was not given; is the energy E (Fig. 3(d)) and E_total (Fig.5) the same as Phi?
- no definition is given for the dipole moment p, which is most likely calculated for dimensionless magnetization; it is not clear how to interpret the ratio p'/p.
3) Numerical simulations:
Information on micromagnetic simulations is laconic. The authors should clarify:
- how they chose the initial configuration for relaxation,
- for what material parameters and for what field the simulation was performed in MuMax3, and whether these values are realistically?
- what is the thickness of the layer?
4) System stability analysis
- the role of the individual contributions to the total energies is not disusssed. The authors consider separately only the contribution of the DMI interaction. Is this contribution decisive for the formation of the total energy minimum for the bi-meron beam? What is the role of the other contributions: exchange energies, anisotropy energies and Zemman energy?
- It may also be interesting to show how these energies vary as a function of both the distance of the bimerons and the angles formed by the dipole moments of the two bimerons relative to the axes on which they are aligned.
4) Bibliography
- For some works titles are not provided
- Please consider citing these papers: https://doi.org/10.1103/PhysRevB.99.060407, https://doi.org/10.1103/PhysRevB.103.174416
Author Response
Kindly see the attached file

Reviewer 3 Report
Comments and Suggestions for Authors
The paper of Mukai and Leonov is a timely and very interesting contribution to the realm of skyrmionics and general noncolinear spin textures in chiral magnets. The authors provide an exceptionally thorough analysis of bimerons and their (attractive) interaction, stemming from their internal structure and resulting in a rich variety of their cluster configurations or even widely interconnected networks (dubbed ‘polymers’ by the authors).
The paper is written well, figures are clear and prepared with care. As a minor remark, several paragraphs in the paper begin with a misplaced ‘t’.
In my opinion, the introduction is unnecessarily broken in its flow by the theoretical field description of skyrmions/merons/bimerons, which may entirely be shifted to a separate theory section.
Generally speaking, the discussion of different topological structures and their interaction may benefit from comparison to the field of superconductivity, or at least citing the articles that consider similar topological entities in superconductors and superfluids (for the broader interest of the readership; see e.g. works on skyrmions in p-wave superconductors, multicomponent superconductors, vortex-antivortex molecules, skyrmion lattices in multicomponent BECs, etc).
Regarding the stability of different skyrmionic textures (in this case bimeronic), one should certainly cite the works of Besarab, also Stosic, employing the GNEB method. For attractive skyrmion-skyrmion interactions, one should cite Rozsa et al and their PRL from 2016. Obviously, recent review on skyrmion dynamics and pinning, by Reichhardt et al in RMP 2022 should be cited accordingly. Regarding the use of MuMax3, the authors should also cite the more recent review of its applications, in DOI: 10.1088/1361-6463/aaab1c (J. Phys. D-Appl. Phys. 2018).
Regarding the physics of the article, I find it important to discern it from mathematics. In particular, it should be explained better what exactly determines the optimal bimeron-bimeron distance (minimum of the interaction), and how that distance compares to the characteristic lengthscales of magnetic exchange. It would also be of relevance to characterize the ‘many-body’ contribution to that interaction, i.e. plot an interaction as a function of distance for a cluster of bimerons, and show how different it is from the sum of pairwise interactions shown in Figure 2.
Finally, the authors identify a very interesting transition from skyrmion lattice to bimeron networks, which would benefit from the energy barrier analysis between them (and thereby improve understanding of the lifetime and temperature stability of such structures).
With these improvements, I can recommend this paper for publication, as it contains a wealth of novel information that will stimulate further theoretical and experimental work in the subject.
Author Response
Kindly see the attached file

Round 2
Reviewer 3 Report
Comments and Suggestions for Authors
The authors have made an effort to accommodate the remarks of the reviewers. The present version of the manuscript is improved, and may be accepted for publication.
That said, it remains disappointing that the authors did not provide any of the energy barriers that would support the stability of the reported novel spin textures. I understand that this is not an easy task, but I deem it very necessary for this type of a study. I expect that the authors will live up to their response and perform such a study in a follow-up paper.
Related to the above, I did not fully agree with the statements in the introduction about the 'topological protection' of skyrmions. This is indeed true, for continuous magnetization fields, but not for discrete atomistic systems. Skyrmions may well collapse on themselves, especially in presence of disorder and pinning, as discussed by Muckel et al experimentally, and Stosic et al theoretically. I would ask the authors to polish the related statements accordingly.
On the other hand, the added paragraph on skyrmions in superconductors is a bit particular on the work of Babaev et al (there are also several other relevant works, with 'skyrmions' even in the title), whereas it should have been more generic and encompassing different condensed matter systems where topological skyrmion objects can be found in the pseudospin textures (specifically defined for that system). In that sense, multicomponent Bose-Einstein condensates are also a good example. I hope authors can still polish that part of the text too.
But seen overall, the paper merits publication in Nanomaterials.
